# Ligand engineering enhances (photo)electrocatalytic activity and stability of zeolitic imidazolate frameworks via in-situ surface reconstruction

Zheao Huang [1], Zhouzhou Wang [2], Hannah Rabl [1], Shaghayegh Naghdi [1], Qiancheng Zhou[2], Sabine Schwarz[3], Dogukan Hazar Apaydin[1], Ying Yu[2] ✉ & Dominik Eder [1] ✉

The current limitations in utilizing metal-organic frameworks for (photo) electrochemical applications stem from their diminished electrochemical stability. In our study, we illustrate a method to bolster the activity and stability of (photo)electrocatalytically active metal-organic frameworks through ligand engineering. We synthesize four distinct mixed-ligand versions of zeolitic imidazolate framework-67, and conduct a comprehensive investigation into the structural evolution and self-reconstruction during electrocatalytic oxygen evolution reactions. In contrast to the conventional single-ligand ZIF, where the framework undergoes a complete transformation into CoOOH via a stepwise oxidation, the ligand-engineered zeolitic imidazolate frameworks manage to preserve the fundamental framework structure by in-situ forming a protective cobalt (oxy)hydroxide layer on the surface. This surface reconstruction facilitates both conductivity and catalytic activity by one order of magnitude and considerably enhances the (photo)electrochemical stability. This work highlights the vital role of ligand engineering for designing advanced and stable metal-organic frameworks for photo- and electrocatalysis.

Metal-organic frameworks (MOFs) are intricate structures comprised of secondary building units (SBUs) or clusters formed by the coordination of metal nodes with organic ligands, the synthesis and applications of which have been thoroughly investigated within the materials field[1,2]. A surge in research efforts has explored the direct applications of MOFs in electrochemical applications, where the metal nodes often serve as active catalytic sites for electrocatalysis, especially in Co-MOFs[3,4], Ni-MOFs[5–7], and mixed-metal MOFs[8,9]. Despite these advancements, the stability and conductivity of MOF materials in electrochemical environments and their feasibility as direct electrocatalysts remain contentious due to the relatively weaker coordinating

bonds between metal nodes and ligands, in contrast to the robust ionic bonds found in traditional inorganic materials[10,11].

For instance, the direct electrocatalysis of pristine zeolitic imidazolate framework-67 (ZIF-67) leads to the complete reconstruction (CR) of its organic framework over time and, eventually to phase transition to $Co(OH)_2$ and CoOOH in alkaline environments (KOH)[12]. Despite the higher catalytic activity of oxygenated cobalt species, the self-reconstruction is typically unavoidable, making it impossible to consider ZIF-67 as an electrocatalyst directly[13,14]. Tang et al. found a two-step dynamic structural reconstruction in NiCo-MOF-74 during the oxygen evolution reaction (OER) process, from the MOF structure

[1]Institute of Materials Chemistry, Technische Universität Wien, 1060 Vienna, Austria. [2]Institute of Nanoscience and Nanotechnology, College of Physical Science and Technology, Central China Normal University, 430079 Wuhan, China. [3]Service Center for Electron Microscopy (USTEM), Technische Universität Wien, 1040 Vienna, Austria. ✉e-mail: yuying01@ccnu.edu.cn; dominik.eder@tuwien.ac.at

to NiCo(OH)$_2$ and then further to NiCoOOH at higher potentials[15]. This nature is prevalent when MOFs are used as OER anodes in alkaline electrolytes, where they undergo the irreversible electro-oxidation process[16,17]. Ultimately, MOFs are constructed into metal-based oxyhydroxides as the real catalytic species, rather than the original metal node. The fundamental nature of the CR process of MOFs during electrochemical treatment is the ligand substitution process, which disrupts the framework stabilization and primitive metal nodes of MOFs. Therefore, we contend that in addition to enhancing electrocatalytic activity, improving the electrochemical stability and mitigating the CR process of MOFs is particularly crucial in (photo) electrochemical applications.

Designing stable MOF electrocatalysts through ligand engineering has gained increasing attention[18,19]. For instance, Dang et al. introduced multiple ligands in MOFs, constructing a multivariate MOF-5 structure with unique properties of CO$_2$-selective adsorption[20]. ZIF-62, which incorporates two different ligands, exhibits suitable thermal stability and undergoes melting before thermal decomposition, making it widely employed in quenching to form distinct MOF glasses[21–23]. Others, Wang et al. and Wu et al., prepared a series of the MOFs or MOF composites as stable and efficient photocatalysts through ligand engineering[24,25]. We thus envision the development of a stable electrocatalyst by employing ligand engineering on ZIFs to balance the electrochemical stability and activity.

According to the hard-soft acid-base (HSAB) principle, two strategies can be envisioned to construct stable MOFs based on the coordination bond strength: (a) hard acid/hard base; (b) soft acid/soft base[25,26]. Unfortunately, commonly used low-valent metals (Zn$^{2+}$ and Co$^{2+}$) in ZIFs belong to soft acids, and it is difficult to design ZIFs with suitable crystallinity using corresponding soft bases (such as pyrazolate ligand)[27]. Considering the literature report claiming to enhance the electrical conductivity of MOFs by tuning the stacked aromatic carbon rings[28], and combining with our previous studies on mixed-ligand ZIFs[29], we here select four secondary ligands featuring aromatic carbon ring and amino group. These ligands were mixed with the original ligand by microwave synthesis reactor, thereby constructing ligand-engineered ZIFs (LE-ZIFs) aimed at achieving a balance between electrochemical activity and stability. Previous studies have easily confused the electrochemical "performance" stability and electrochemical stability, and the main reason for this ambiguity lies in the absence of in-situ electrochemical characterization[10,30,31]. Here we combine the continuous cyclic voltammetry (CV), in-situ UV–Vis absorption spectroscopy, in-situ Raman spectroscopy and in-situ photoluminescence (PL) spectroscopy to elucidate the reconstruction phenomena of as-prepared ZIFs during (photo)electrocatalytic reaction. Ex-situ characterizations and density functional theory (DFT) calculations provide compelling evidence for the robust electrochemical stability and OER activity of AE-ZIF (refer to Fig. 1a, mixed 2-

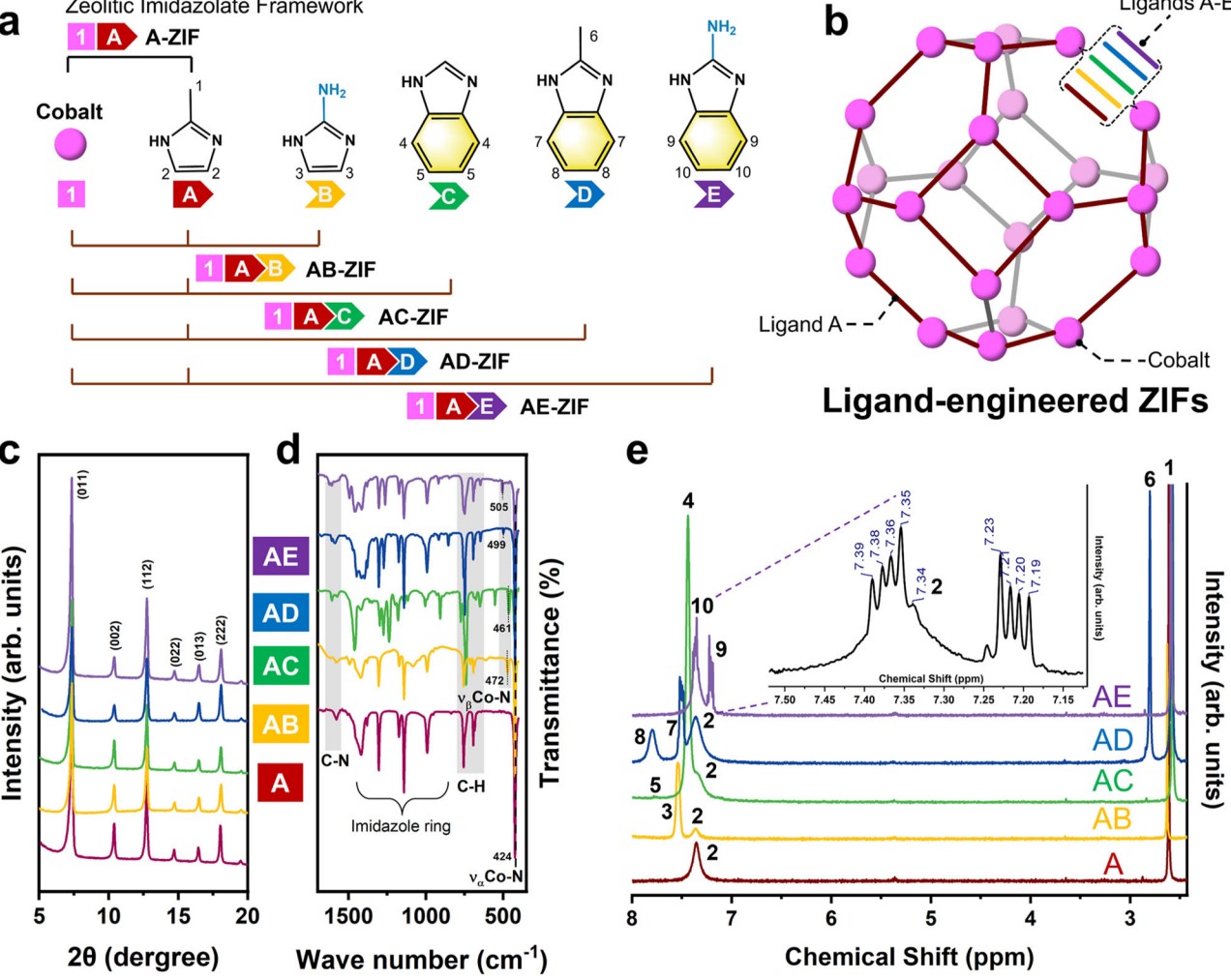

**Fig. 1 | Structural characterization of Ligand-engineered ZIFs. a, b** Schematic diagram of the various ligands A-E mixed in LE-ZIFs (**a**) and unit cell of each LE-ZIFs (**b**). **c–e** XRD patterns (**c**), ATR-IR (**d**) and ¹H NMR spectroscopy (**e**) of A-ZIF, AB-ZIF, AC-ZIF, AD-ZIF and AE-ZIF, more details in Figs. S1, S3 and S21–S25. All samples were digested using d⁴-acetic acid and then tested by ¹H NMR. The numbers in Fig. 1e are for H at the corresponding positions on the ligand in Fig. 1a.

aminobenzimidazole). This study opens up a feasible pathway to enhance OER activity and stability by modulating the organic ligands in the stabilized LE-ZIFs synthesis.

## Results

### Structural characterization of Ligand-engineered ZIFs

Single-ligand ZIF-67 (A-ZIF) is formed by the tetrahedral coordination of cobalt ions with the ligand 2-methylimidazole (2-mIm, termed as A) which creates a sodalite (SOD) structure. For the ligand-engineered ZIFs (LE-ZIFs), we introduced four secondary ligands: 1H-Imidazol-2-amine ($NH_2$-mIm, B), Benzimidazole (bIm, C), 2-methylbenzimidazole (2-bIm, D), and 2-aminobenzimidazole ($NH_2$-bIm, E). Each ligand was mixed with the original ligand A using a microwave process to construct the ZIF structures (Fig. 1a, b). The X-ray powder diffraction (XRD) pattern and attenuated total reflection infra-red (ATR-IR) spectrum of A-ZIF match well with the previous literature reports[32,33]. The XRD patterns of the LE-ZIFs also exhibit a close resemblance to those of A-ZIF, indicating that the intrinsic structure is well-preserved and exhibits high crystallinity (Figs. 1c and S1). The absence of secondary or impurity phases implies that the ligand substitution occurred solely within the framework. Note that the embedding of ligand B results in structural distortion in AB-ZIF, which impacts the "optimal" particle growth, e.g., an increase in particle size (as observed in scanning electron microscopy, SEM, in Fig. S2).

ATR-IR spectra reveal the incorporation of secondary ligands within the LE-ZIFs framework. All as-prepared ZIFs show the characteristic band about 435 cm⁻¹, which corresponds to the $Co$-$N_\alpha$ coordination with ligand A. New bands between 472 and 505 cm⁻¹ correspond to $Co$-$N_\beta$ vibrations with the respective secondary ligands (Figs. 1d and S3). The simultaneous presence of these two metal-ligand vibrational modes confirms the successful incorporation of both ligands in the LE-ZIFs framework.

$N_2$ physisorption at 77 K reveals that A-ZIF exhibits a high specific surface area (1867.9 m² g⁻¹) and a 1.2 nm microporous structure (0.655 cm³ g⁻¹ in Table S1). In comparison, the introduction of a secondary ligand has reduced the respective specific surface areas (717.8–1092.9 m² g⁻¹) and pore volumes (0.365-0.489 cm³ g⁻¹), while preserving the microporous structure around 1.2 nm (Figs. S4 and S5).

¹H nuclear magnetic resonance (¹H NMR) spectrum of A-ZIF contains the expected features (peaks 1 and 2) corresponding to the methyl group and imidazole. In addition, the spectra of LE-ZIFs (Fig. 1e) exhibit new features (peaks 3–10) that correspond to the respective secondary ligands B-E. Quantitative analysis of the ligand composition by ¹H NMR reveals that the actual ratios are as follows: ligand B at 51.2 ± 1.6 mol%, ligand C at 41.4 ± 0.4 mol%, ligand D at 39.0 ± 0.7 mol%, and ligand E at 36.9 ± 1.2 mol%. Their respective contents are in good agreement with the nominal values (as detailed in Table S3), providing further validation of the precision in ligand incorporation into the LE-ZIFs.

### Electrocatalytic studies

For all electrocatalytic studies, as shown in Fig. S6a, the as-prepared ZIFs were drop-cast onto fluorine-doped tin oxide (FTO) glass and used as working electrodes in an aqueous 1 M KOH solution, with the potential referenced to the reversible hydrogen electrode (RHE). At a high potential (0.85–1.55 V), the CV plot of A-ZIF changes with increasing cycle numbers and two distinct redox peaks appear denoted as $R_{SL-A}1$ and $R_{SL-A}2$ (Fig. 2a). The $R_{SL-A}1$ represents the electro-oxidation process from $Co^{2+}$ to $Co^{3+}$ transition, while $R_{SL-A}2$ indicates the $Co^{3+}$ to $Co^{4+}$ transition[34]. The presence of these signals indicates the beginning of self-reconstruction, which intensifies with increasing CV cycles[12]. After 50 cycles, the current density of A-ZIF reaches a maximum and then starts to decrease, indicating a partial degradation of the framework, accompanied by the formation of oxygenated Co species. This trend is somewhat delayed at medium potentials

(0.85–1.30 V). At low potentials (0.85–1.05 V), A-ZIF shows typical capacitive behaviour, with the current mainly originating from the adsorption and release of hydroxide ions within the double-layer capacitor (Fig. S6b–e). Other LE samples exhibit distinctly differences CV plots; for instance, AB and AE only show signs of the $Co^{2+}$ to $Co^{3+}$ transition, while the oxidation of $Co^{3+}$ to $Co^{4+}$ seems to be suppressed, which is closely related to their mixed secondary ligands (Fig. S7).

The quantitative analysis of the anodic peaks in the CV plots is summarized in Figs. 2b and S8. Concerning the anodic peak (region 1), the charges of $R_{LE-AE}1$ reaches the highest value (0.047 mC) after 80 cycles, followed by $R_{LE-AC}1$ and $R_{LE-AD}1$, while $R_{SL-A}1$ remains the lowest. This anodic charge corresponds to the number of electron-accessible $Co^{2+}$ sites[12]. Additionally, the anodic peak positions generally shift towards the higher potentials, indicating the transformation of $Co^{2+}$ species into more stable states. The final stabilization of $R_{LE-AE}1$ at a maximum of 1.329 V suggests that AE-ZIF requires a higher potential to be electro-oxidation compared to the other ZIF samples. When using electrochemical double-layer capacitance ($C_{dl}$) to reflect the electrochemical active surface area (ECSA, in Fig. S9), it is evident that after 100 CV cycles, the reduction in ECSA for AE-ZIF (8%) is less pronounced compared to the significantly decrease observed for A-ZIF (48%). This also indicates that AE-ZIF exhibits stronger resistance properties to continuous electro-oxidation.

The variation trend of $O_2$ evolution rates and reaction parameters during CV test is illustrated in Fig. 2c. Oxygen flow data are provided in real-time by an in-situ oxygen flow detection configuration connected to the electrolytic cell. For A-ZIF, the overpotential and Tafel slope reach the minimum at 50 cycles and then increase, exhibiting an inverted volcano-shaped curve. This implies that the optimal OER activity of A-ZIF occurs at 50 cycles, and an additional CV cycle leads to excessive structural reconstruction and degradation, thus decreasing the performance. In contrast, AE-ZIF stabilizes its overpotential and Tafel slope after 80 cycles, exhibiting higher electrocatalytic activity and $O_2$ evolution rate. Compared to other LE samples in Figs. S11 and S14, it can be inferred that AE-ZIF has the highest OER activity and electrical conductivity, as discussed in Supplementary Section S2.

According to the multistep potentiometry, samples A and AE exhibit relative potential fluctuations at certain current densities due to the reconstruction effects (Figs. 2d and S12a). Further combined with CV analysis, applying potentials of 1.05, 1.35 and 1.65 V to detect their time-dependent amperometry is reasonable (Fig. 2e and Fig. S12b, c). Evidently, the current density rapidly increases within the initial 2 h of OER, indicating rapid reconstruction of the ZIFs electrode, consistent with the CV analysis results. Subsequently, the current density and $O_2$ evolution rate of A-ZIF decrease continuously starting after 2.5 h, indicating ongoing reconstruction, ultimately losing 35.4% $O_2$ evolution rate after 12 h at 1.65 V. In contrast, the structural reconstruction in AE-ZIF ceased after 2 h and maintained a more stable current under constant potential or intermittent bias, exhibiting a current density and $O_2$ evolution rate that were an order of magnitude higher than those of A-ZIF (Fig. 2e and Fig. S13d).

Hence, combined with the collective insights from current density, $O_2$ evolution rate, conductivity, electrochemical impedance of other LE samples (Figs. S7–15, refer to Supplementary Section S2), the performance and structural reconstruction under electrocatalysis in as-prepared ZIFs are confirmed. The conclusion drawn is that AE-ZIF outperforms A-ZIF and other LE-ZIFs in both OER activity and electrochemical stability, with the process of self-reconstruction being effectively suppressed.

### Structural evolution and reconstruction during electrocatalytic reaction

Figure 3a shows the disappearance of Bragg peaks belonging to sodalite-type ZIF-67 in the XRD pattern of A-ZIF within the CV cycles in

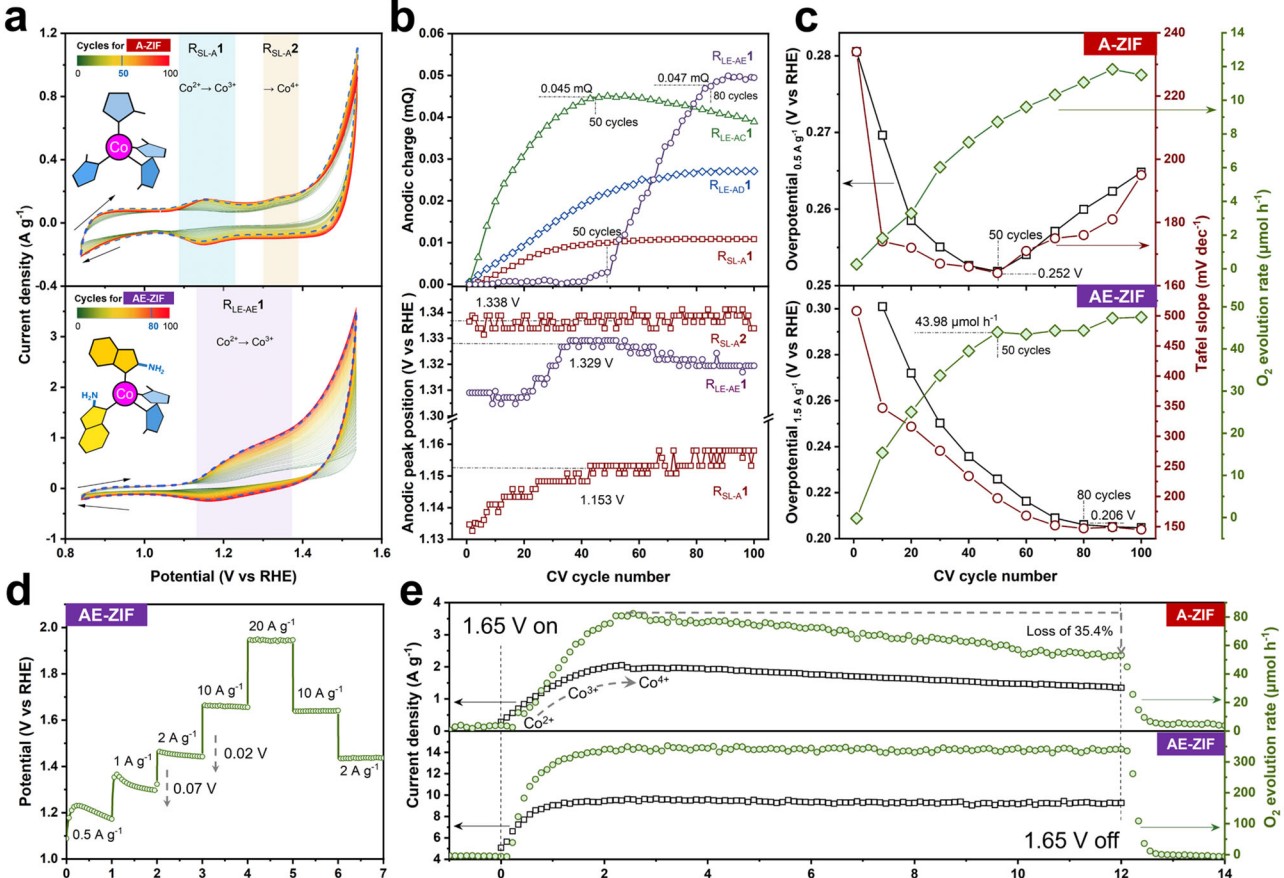

**Fig. 2 | Cyclic voltammetry, potentiometry and amperometry of ZIFs.**
**a** Continuous cyclic voltammograms at high potential window between 0.85 to 1.55 V of A-ZIF and AE-ZIF. Dashed line is the current density maximum, the scan rate is 10 mV s⁻¹. **b** Anodic charge and peak position plots with CV cycles of A-ZIF and LE-ZIFs. **c** Overpotential (current density is 0.5 A g⁻¹ for A-ZIF and 1.5 A g⁻¹ for AE-ZIF), Tafel slope and $O_2$ reaction rate plots with CV cycles of A-ZIF and AE-ZIF.

The calculation data for the Tafel slope are from Fig. S10. **d** Multistep potentiometric plot of AE-ZIF at different current densities. **e** Amperometric and $O_2$ evolution rate plots of A-ZIF and AE-ZIF at certain potentials of 1.65 V. All data were not calibrated with the iR correction, and the measurements were performed only once due continuous running.

the high potential window. At 50 CV cycles, additional peaks at 19.1° and 38.0° appeared that correspond to the (001) and (101) planes of $Co(OH)_2$[35]. Beyond 100 cycles, the $Co(OH)_2$ peaks disappeared while new diffractions arose at 19.9° and 39.1°, corresponding to the (003) and (101) planes of CoOOH, respectively[35,36]. These changes are mitigated to some extent upon limiting the scanning range of the CV to lower and medium potential windows.

In contrast, the well-defined XRD pattern of AE-ZIF was clearly preserved up to about 100 CV cycles, even in the high potential window, indicating superior stability under all used electrocatalytic conditions. On the other hand, AB-ZIF exhibited the lowest stability, characterized by complete amorphization (Fig. 3b). Samples AC and AD were relatively stable, retaining the Bragg peaks of ZIF under a high potential window. Note that AC-ZIF displays additional CoOOH peaks after 100 CV cycles, indicating a higher degree of electro-oxidation compared to AD and AE.

The structural changes were investigated in more detail by ¹H NMR. The spectra (Figs. S21–25) reveal that the ligand 2-mIm in A-ZIF disappeared almost completely upon cycling, indicating substitution of the original ligand. Conversely, AD and AE showed only a minor loss of secondary ligands of about 3–6 mol% (Table S3), the overall framework structure was largely preserved throughout the electrochemical cycling. This is also evident in the ATR-IR spectra taken after the reconstruction process, which still shows the stretching vibrations from Co-N and 2-mIm ligands in AE-ZIF even after 100 CV cycles, while they disappeared in A-ZIF after 50 cycles (Fig. S26).

Figure 3c, d summarize the results from X-ray photoelectron spectroscopy (XPS) analysis of the as-prepared ZIFs before and after electrocatalysis. The Co $2p_{3/2}$ spectra of A-ZIF show two peaks before the reaction, corresponding to $Co^{2+}$ coordinated to N on the ligand (782.09 eV, red) and satellite peak (787.79 eV, blue)[37,38]. After 100 CV cycles in the medium or high potential ranges, a new distinct peak appeared at 779.14 eV (green) typically ascribed to $Co^{3+}$ generated from $Co^{2+}$ oxidation [39–41]. For comparison, no significant evolution of $Co^{3+}$ was observed in A-ZIF in the low potential window. Note that electrochemical CV cycling induced a gradual decrease in the N/Co atomic ratio of A-ZIF from 3.9 – 2.4 at low potential and to zero in medium and high potential windows (Table S2). This supports the aforementioned framework degradation of A-ZIF which starts with Co-N cleavage at certain potentials. Concurrently, the N/O atomic ratio also decreases from 0.5 to zero, which is explained by the substitution of the 2-mIm ligand by hydroxide radicals (·OH) during the complete reconstruction (CR) process. This is further supported by the vanishing N $1s$ signals corresponding to C-N (400.15 eV, blue) and Co-N (398.85 eV, red), respectively.

In contrast, samples AC, AD and AE show no apparent formation of $Co^{3+}$, even in the high potential window. Moreover, both signals in the N $1s$ region remained unaltered (Figs. 3d, S19 and S20). Notably, the N/Co atomic ratio in AE-ZIF still decreased slightly from 4.5 to 3.6 at high potential window (with low and medium potentials being negligible). This suggests that the AE-ZIF electrode may undergo only partial substitution processes, i.e. such as occurring in the surface

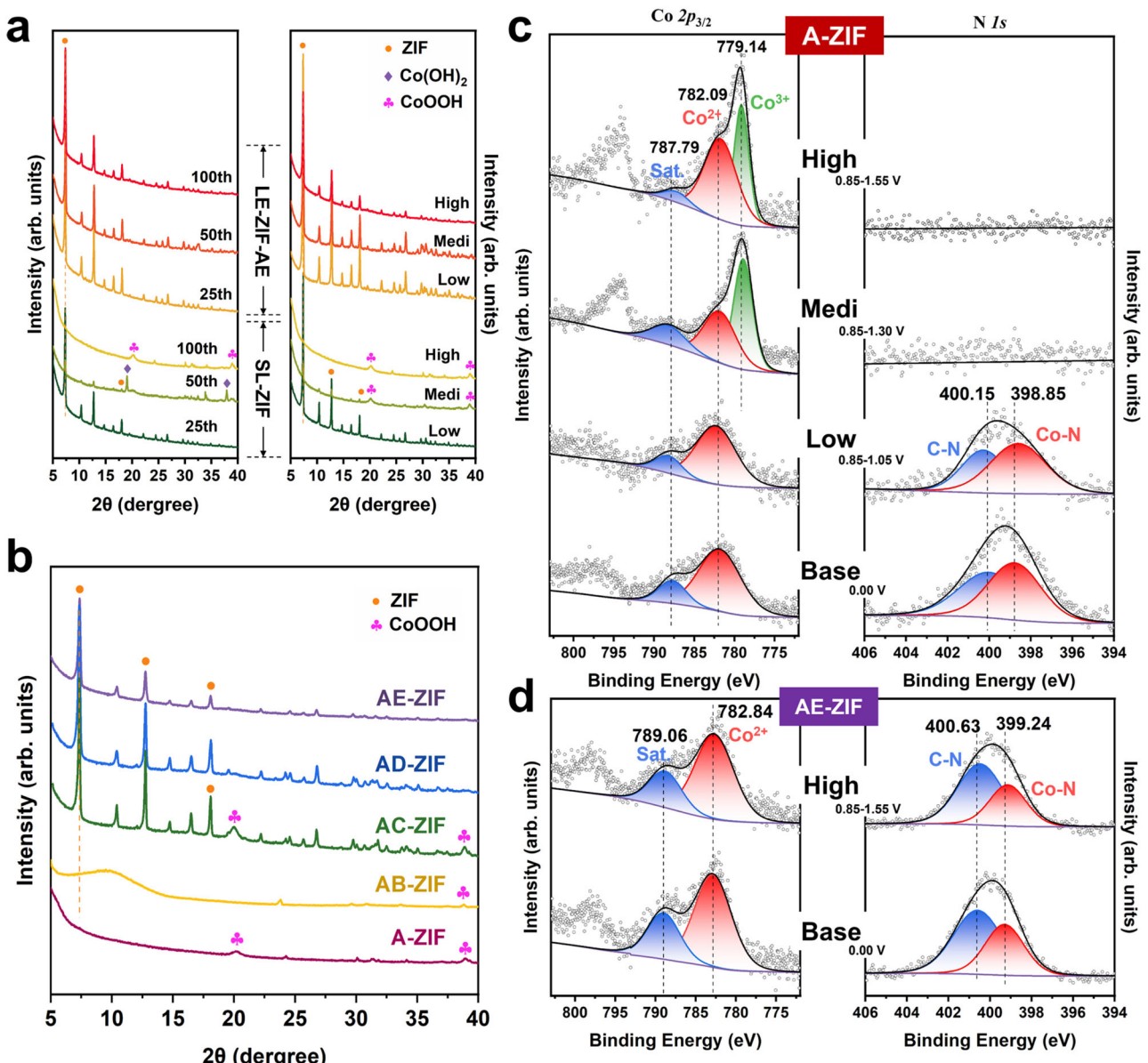

**Fig. 3 | Structure evolution of ZIFs after electrocatalysis. a** XRD patterns of A-ZIF and AE-ZIF with different CV cycles (25, 50 and 100) at 0.85–1.55 V, with 100 CV cycles at different potential windows (Low: 0.85–1.05 V, Medi: 0.85–1.30 V, High: 0.85–1.55 V). **b**, XRD patterns of A-ZIF, AB-ZIF, AC-ZIF, AD-ZIF and AE-ZIF with 100 CV cycles at 0.85–1.55 V. **c**, **d** Co $2p_{3/2}$ and N $1s$ of XPS spectra for A-ZIF (**c**) and AE-ZIF (**d**) with 100 CV cycles at different potential window (Low: 0.85–1.05 V, Medi: 0.85–1.30 V, High: 0.85–1.55 V), for whole sample scan see Fig. S17.

reconstruction (SR) process rather than the no-reconstruction (NR) process. The absence of new signals in the Co $2p_{3/2}$ spectra may be attributed to the incomplete detection caused by the low content of oxygenated Co species in AE-ZIF after the SR process. As predicted, the disappearance of XPS signals in AB-ZIF indicates the degradation of framework structure (Fig. S18), as discussed in Supplementary Section S3.

Transmission electron microscopy (TEM) and energy-dispersive spectrum (EDS) provide more insights into the structural reconstruction and elemental composition of each ZIF particles after 12 h amperometry. In accordance with predictions from XRD and XPS analysis, A-ZIF fails to maintain the dodecahedral morphology and crystal structure post-reaction, as illustrated in Figs. S27 and S28. The morphology ultimately transforms into nanosheets with lattice fringes characteristic of $Co(OH)_2$ and CoOOH, as confirmed by the bright spotting in the selected-area electron diffraction (SAED) pattern[42,43]. The XRD pattern of post-reaction also confirms the presence of high-

valence cobalt oxides (Fig. S16). Elemental mapping through EDS reveals a homogeneous distribution of Co and O elements within the nanosheets as the dominant elements (Fig. S28d), confirming the conversion into (oxy)hydroxide during amperometry.

In contrast, the TEM images of AE-ZIF pre- and post-reaction clearly showcase the particles retain their typical dodecahedral morphology. Importantly, after amperometry, the edges of the AE-ZIF particle are no longer sharp, and the presence of a distinct thin layer suggests the formation of a core-shell structure (Fig. 4a–e). Elemental mapping reveals a noticeable decrease in N and C and a significant increase in Co and O concentration in the shell region, contrasting the homogeneous distribution of each element observed pre-reaction (Figs. 4c and f). The bright spots in the corresponding selected-area fast Fourier transform (FFT) patterns and lattice fringes further confirm the crystalline characters of the outer layer as cobalt (oxy)hydroxide (Figs. 4e and S30). These results suggest that in-situ reconstruction occurs exclusively on the outermost surface of AE-ZIF, eventually

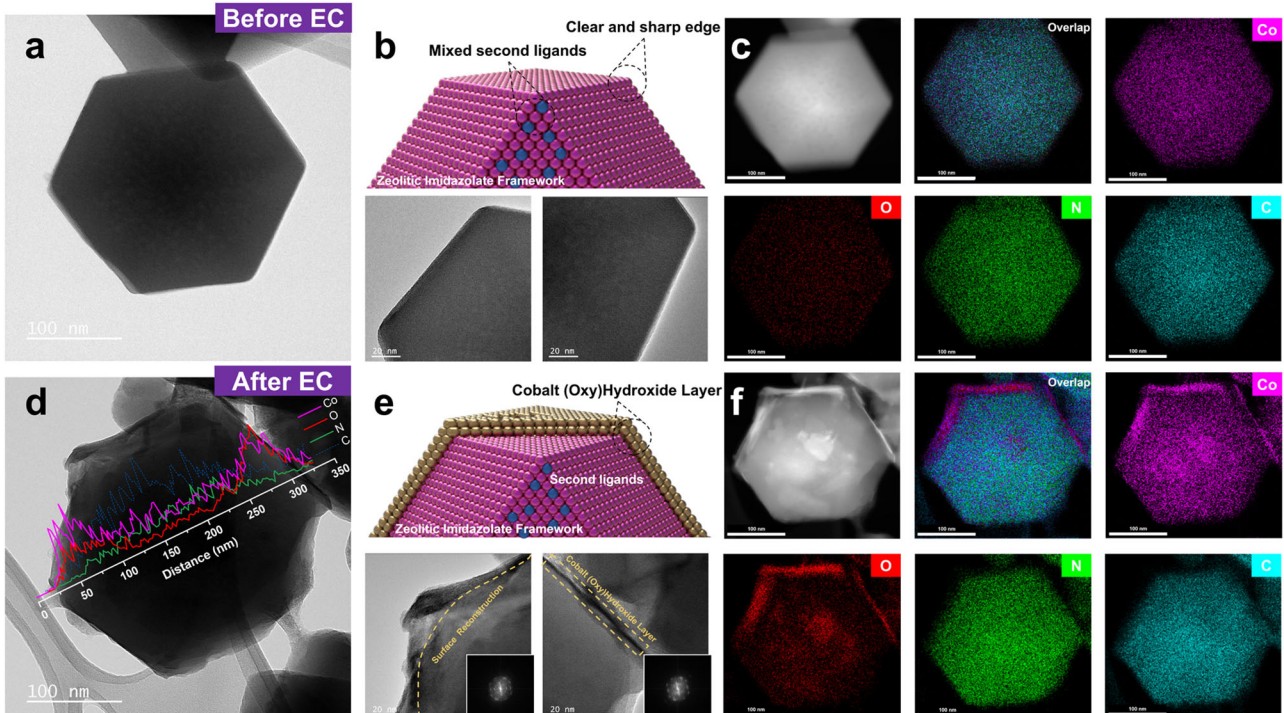

**Fig. 4 | Morphology evolution of AE-ZIF after electrocatalysis. a, d** TEM images of single-particle AE-ZIF before (**a**) and after 12 h amperometry (**d**). **b, e** Schematic diagrams and enlarged TEM images of AE-ZIF edge before (**b**) and after 12 h amperometry (**e**), the lower corresponding FFT patterns. **c, f** Elemental mapping of single-particle AE-ZIF before (**c**) and after (**f**) 12 h amperometry showing the Co, O, N and C distribution.

yielding a thin stable (oxy)hydroxide layer without affecting the core structure. Morphological and elemental composition of other AC and AD after the amperometry also reveals similar core-shell structures through TEM and EDS analysis (Figs. S27–33, refer to Supplementary Section S4).

The combined results clearly indicate that the in-situ formation of a cobalt (oxy)hydroxide layer on the particle surface effectively suppresses the self-reconstruction process, contributing to the outstanding electrochemical stability and OER activity in AE-ZIF.

**In-situ electrochemical spectroscopy analysis**
In-situ electrochemical UV–Vis absorption and Raman spectroscopy was used to gain a clearer understanding of the underlying electro-oxidation process. Similar to the electrocatalytic studies mentioned above, a three-electrode configuration was employed for all electrochemical cells, with ZIFs drop-casted onto the working electrode using the same approach (Figs. 5a and S34). The working electrode remains an FTO glass to allow penetration of UV–Vis and fluorescence light, enabling the reception of structural signals by the spectra receiver.

Figure 5b compares the UV–Vis absorption trends of A and AE during 100 CV cycles, while the other LE samples are shown in Fig. S35a–c. Sample A-ZIF exhibits a distinctive broad signal in the 500−650 nm range, which can be fitted to 539, 567 and 589 nm as spin-orbital coupling triplet peaks, representing the transition of tetrahedral Co species from $^4A_2(F)$ to $^4T_1(P)$[32]. With increasing CV cycles, the intensity of the triplet peaks in A-ZIF rapidly decreases at high potential window, completely disappearing after 40 cycles. This indicates the degradation of the Co-$(mIm)_4$ tetrahedral structure, which is mitigated in the low and medium potential windows (Fig. S35d, e). The broad signal associated with tetrahedral Co species in AE-ZIF remains nearly unchanged, indicating higher electrochemical stability, consistent with the conclusions drawn from our previous electrocatalytic and ex-situ characterization analyses.

In-situ electrochemical Raman spectroscopy provides further sensitivity to unveil the reconstruction of as-prepared ZIFs during electrochemical treatments. The characteristic peaks at 126 − 313 cm$^{-1}$ and 425 cm$^{-1}$ are typically assigned to the vibrational modes of the stretching vibrations of ligand 2-mIm and Co-N bonds within sodalite-type ZIF-67 (Fig. 5c)[44–46]. As the applied potential gradually increases, the intensity of ligand signals in A-ZIF decreases and eventually disappears at 1.40 V. Commencing from 1.20 V, new peaks appear at 498, 535, 586 and 632 cm$^{-1}$, which can be assigned to the vibration modes of Co(OH)$_2$ and CoOOH, respectively[47–49]. Furthermore, the peak at 801 cm$^{-1}$ can be attributed to surface-adsorbed $^*$O-OH species by O-O stretch[50]. Subsequent to 1.55 V, the peak intensity of Co(OH)$_2$ nearly disappears, while the CoOOH peak remains unaltered, corresponding to electro-oxidation process Co$^{2+}$→Co$^{3+}$→Co$^{4+}$ within A-ZIF derived from previous experiments.

In contrast, AE-ZIF shows only weak peaks of Co(OH)$_2$ (498 cm$^{-1}$) and CoOOH (617 cm$^{-1}$) after 1.35 V and, importantly, no discernible changes of the inherent ligand signals. Note that the CoOOH peak in AE-ZIF is comparatively weak and broad, indicating its confinement to the surface of the particles, which also accounts for the undetectability of Co$^{3+}$ signal in XPS analysis. The appearance of (oxy)hydroxide signals is thus related to the in-situ formation of a protective thin layer on the AE-ZIF surface during the SR process, in line with the CV and TEM results (Figs. 2 and 4). The decrease in peak intensity after 1.60 V is attributed to the O$_2$ evolution of gas bubbles on the FTO electrode surface, affecting the signal reception of the Raman laser. Similarly, continuous CV tests show the CR in A-ZIF starting at 20 cycles, whereas SR in AE-ZIF is postponed, indicating mixed-ligand AE-ZIF has superior stability (Fig. 5d). For other LE samples, the ligand signals are also well-preserved in the AC and AD, whereas in AB-ZIF, they rapidly disappear as the reaction progresses (Figs. S36 and S37, refer to Supplementary Section S5).

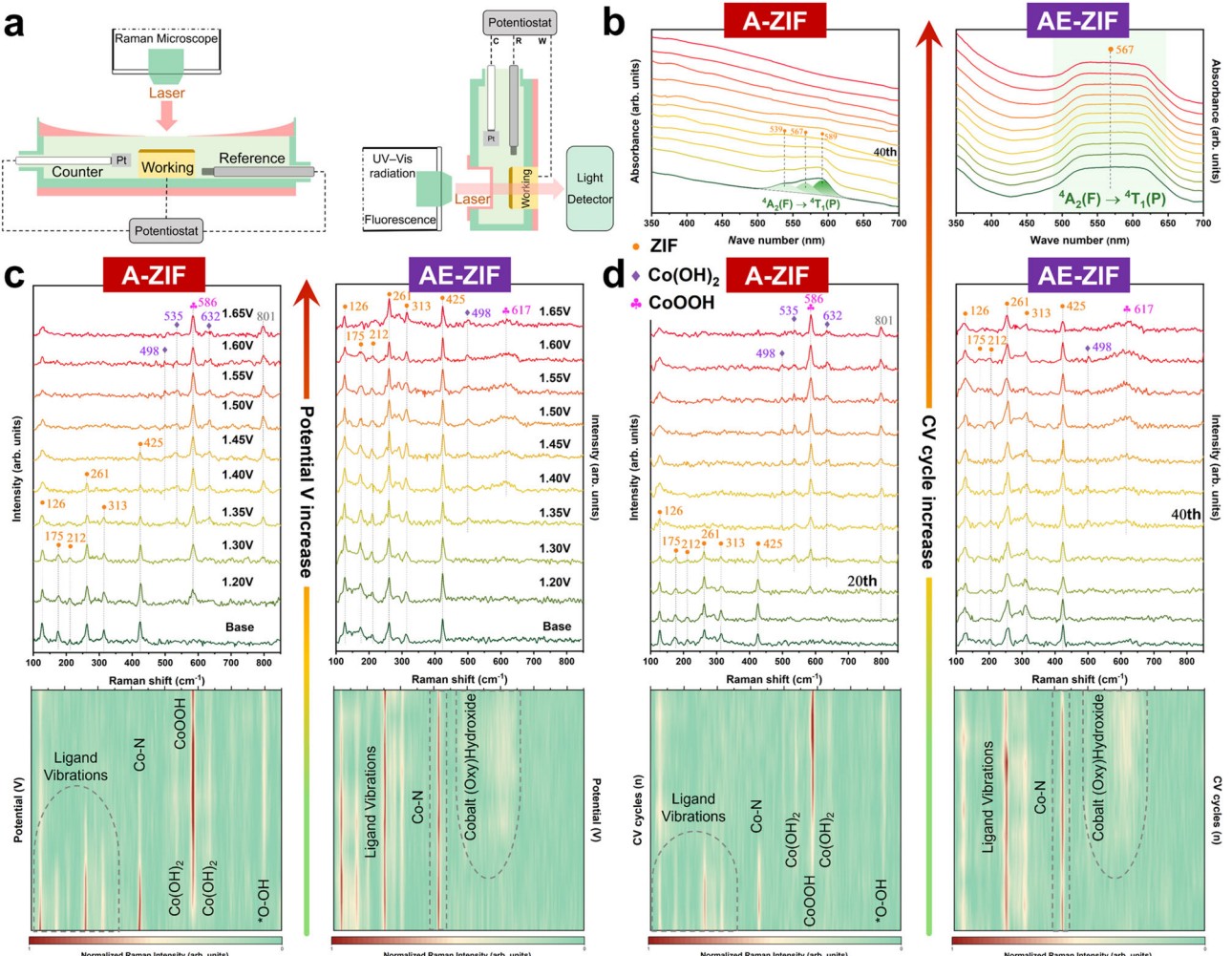

**Fig. 5 | Structural reconstruction of as-prepared ZIFs by in-situ spectro-electrochemical analysis. a** Cell schematic diagram in the three-electrode configuration of in-situ electrochemical Raman and UV–vis absorption spectroscopy. **b** In-situ electrochemical UV–Vis absorption spectroscopy of A-ZIF and AE-ZIF with 100 CV cycles at 0.85–1.55 V. **c** In-situ electrochemical Raman spectroscopy and 2D contour plots of A-ZIF and AE-ZIF at various applied potentials from 1.20 – 1.65 V. **d** In-situ electrochemical Raman spectroscopy and 2D contour plots of A-ZIF and AE-ZIF with 100 CV cycles at 0.85–1.55 V.

## Comprehensive analysis of photoelectrocatalytic activity, photoluminescence, and band gaps

In the field of light-absorbing materials, Co-based ZIFs have garnered attention for their inherent capability to absorb visible light. Integrating electro- with photocatalysis, referred to as photoelectrocatalysis, presents a promising avenue for augmenting the OER activity of ZIFs. Moreover, the imposition of an applied potential serves as an effective strategy to prevent the recombination of photogenerated electron-hole (e−/h) pairs[51]. As anticipated, samples A and AE demonstrate visible light photoelectrocatalytic capabilities; however, their current density and $O_2$ evolution rate have markedly different variation trends due to the reconstruction effect (Fig. 6).

Figure 6a illustrates the outcomes of in-situ electrochemical photoluminescence (PL) analysis. The inherent PL peak of A-ZIF are clearly blue-shifted from 566 to 482 nm after 40 CV cycles, showcasing the phase transition by the CR process, consequently altering its fluorescence properties[52]. Similarly, the band gaps obtained through UV–vis absorption spectroscopy and Mott–Schottky plots[53] exhibit a consistent trend, increasing from 1.75 eV at the beginning to 2.75 eV after 40 CV cycles accompanied by a significant gap widening (Fig. 6b and Fig. S38). Current density and $O_2$ evolution rate both increase upon visible light exposure at the early stages, but decrease after only

2 h, eventually (after 6 h) approaching values similar to those obtained under dark conditions (Fig. 6c). This phenomenon is clearly linked to the CR process in A-ZIF during electrocatalysis, which leads to a loss of the inherent visible light absorption capability along with an increase in band gap.

In contrast, current density and $O_2$ evolution rate of AE-ZIF remain highly stable upon illumination throughout 12 h amperometry, which underlines that the impact of light exposure on the structural and catalytic stability is negligible (Fig. 6c). Notably, within the initial hour of amperometry, the AE-ZIF even shows a rapid rise in efficiency during in-situ formation of cobalt (oxy)hydroxide layer through the SR process. Subsequently, the efficiency stabilizes before it eventually experiences a comparatively small decrease of only 7.4% after 12 h. Note that under illumination, AE-ZIF yields a stable current density of 16.3 A g−1 and an $O_2$ evolution rate of 570.8 μmol h−1, which is twice that observed under dark conditions (Fig. 2e). The stable performance of AE-ZIF in contrast to A-ZIF can further be linked to the band gap and PL intensity, which experience only a small decrease from 1.35 to 1.21 eV, as only surface reconstruction occurs (Fig. 6d, e and Fig. S39). Samples AC and AD exhibit visible-light catalytic activity, whereas the AB-ZIF has none, primarily attributed to an excessively wide band gap (Figs. S38–45, refer to Supplementary Section S6).

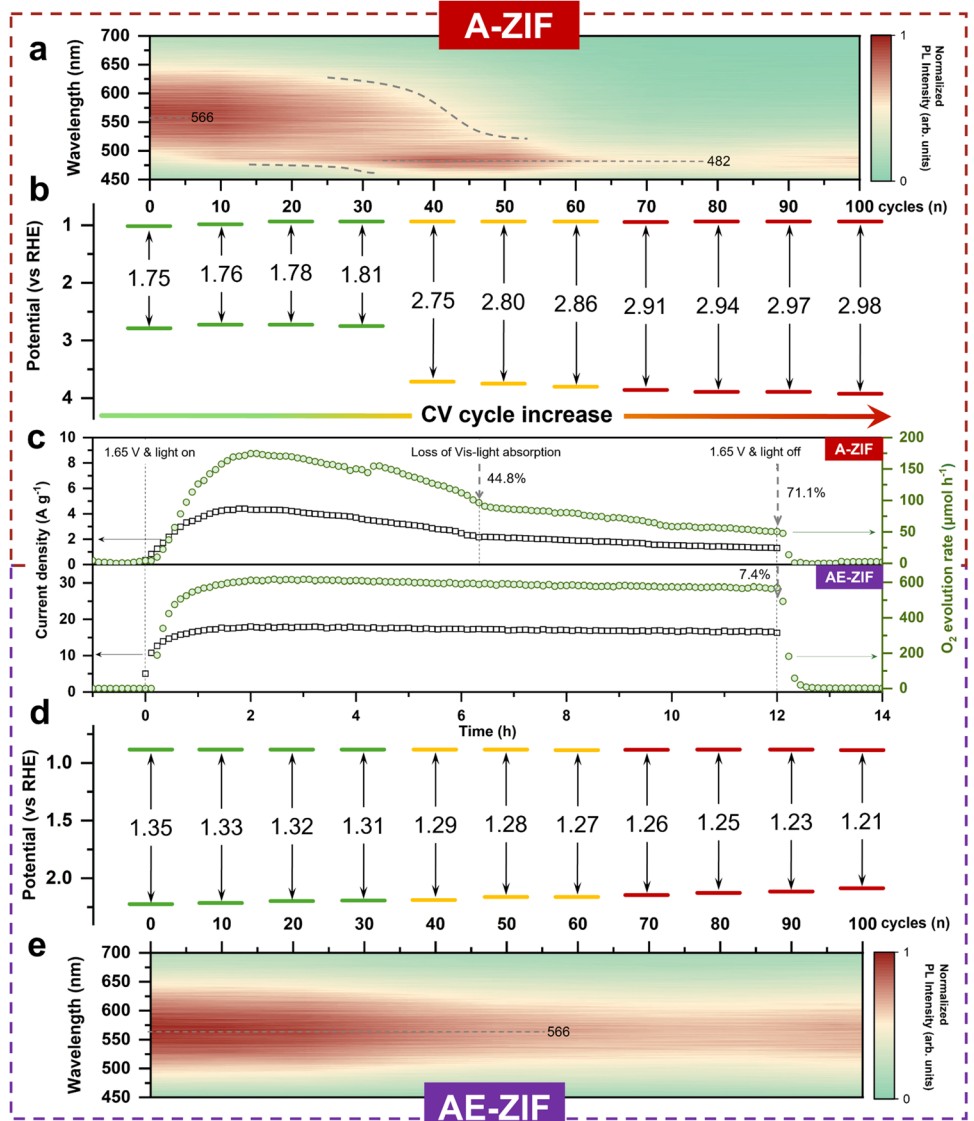

**Fig. 6 | Photoelectrocatalytic OER activity, PL intensity and band gaps. a, b** In-situ electrochemical Photoluminescence spectroscopy (**a**) and band potential diagram (**b**) of A-ZIF with 100 CV cycles at 0.85–1.55 V. **c** Amperometric plots and O₂ evolution rates of A-ZIF and AE-ZIF at certain potentials of 1.65 V and visible light. **d, e** Band potential diagram (**d**) and in-situ electrochemical PL spectroscopy (**e**) of AE-ZIF with 100 CV cycles at 0.85–1.55 V.

## DFT simulation and Mechanistic Study of Ligand-engineered ZIFs as (photo)electrocatalysts

In the preceding analysis, samples A and AB undergo the CR process, leading to the formation of additional phases, i.e. (oxy)hydroxide nanosheets in A-ZIF and an amorphous phase in AB-ZIF. This is accompanied by a considerable change in surface area, band gap, morphology and OER activity. Conversely, samples AC, AD and AE undergo the SR process within the initial 2 h reaction, forming a cobalt (oxy)hydroxide layer on the outermost shell without compromising the internal framework structure. This surface limitation preserves their electrochemical stability during 12 h amperometry, as well as their optoelectronic properties and band gap. Among these samples, AE-ZIF demonstrated the highest OER performance and stability.

To elucidate the fundamental reasons for the activity-stability differences among various LE-ZIFs in electrocatalysis, density functional theory (DFT) calculations were performed using optimized models with about 30% mixed-ligands (Fig. 7a). The density of states (DOS) for LE-ZIF models in Fig. 7b reveal that the upper spins of AC, AD and AE are closer to the Fermi level compared to A and AB, indicating

their electron structures likely are more conductive, which aligns well with the experimental results. This can be attributed to the π-π stacking aromatic carbon rings from the secondary ligands (C, D and E) in the crystal as conductive pathway, enhancing the electrical conductivity[28,54]. Furthermore, the energy difference (ΔE) between $\varepsilon_d$ (Co band center) and $\varepsilon_p$ (N band center) of AE-ZIF is the lowest at 1.33 eV, followed by AD-ZIF at 1.45 eV, while AB-ZIF exhibits the highest at 4.56 eV (Fig. S51 and Table S4). According to previous studies[55,56], this suggests that the Co-N in AE-ZIF has the strongest orbital hybridization and greatest covalency, facilitating electron transfer between Co and N atoms. Similarly, the additional electron adsorption of the -NH₂ group in ligand E also contributes favourably to electrocatalytic reaction, and thereby improving the overall OER performance[57,58].

Further DFT calculations were performed by connecting ligands A and E into the CoOOH model. The charge density difference diagram in Fig. 7c shows that ligand E exhibits a more concentrated electron density at the Co-N-O interface compared to original ligand A. Moreover, the Gibbs free energy was also calculated for each elementary step for OER, using the adsorbate evolution mechanism (AEM). Upon

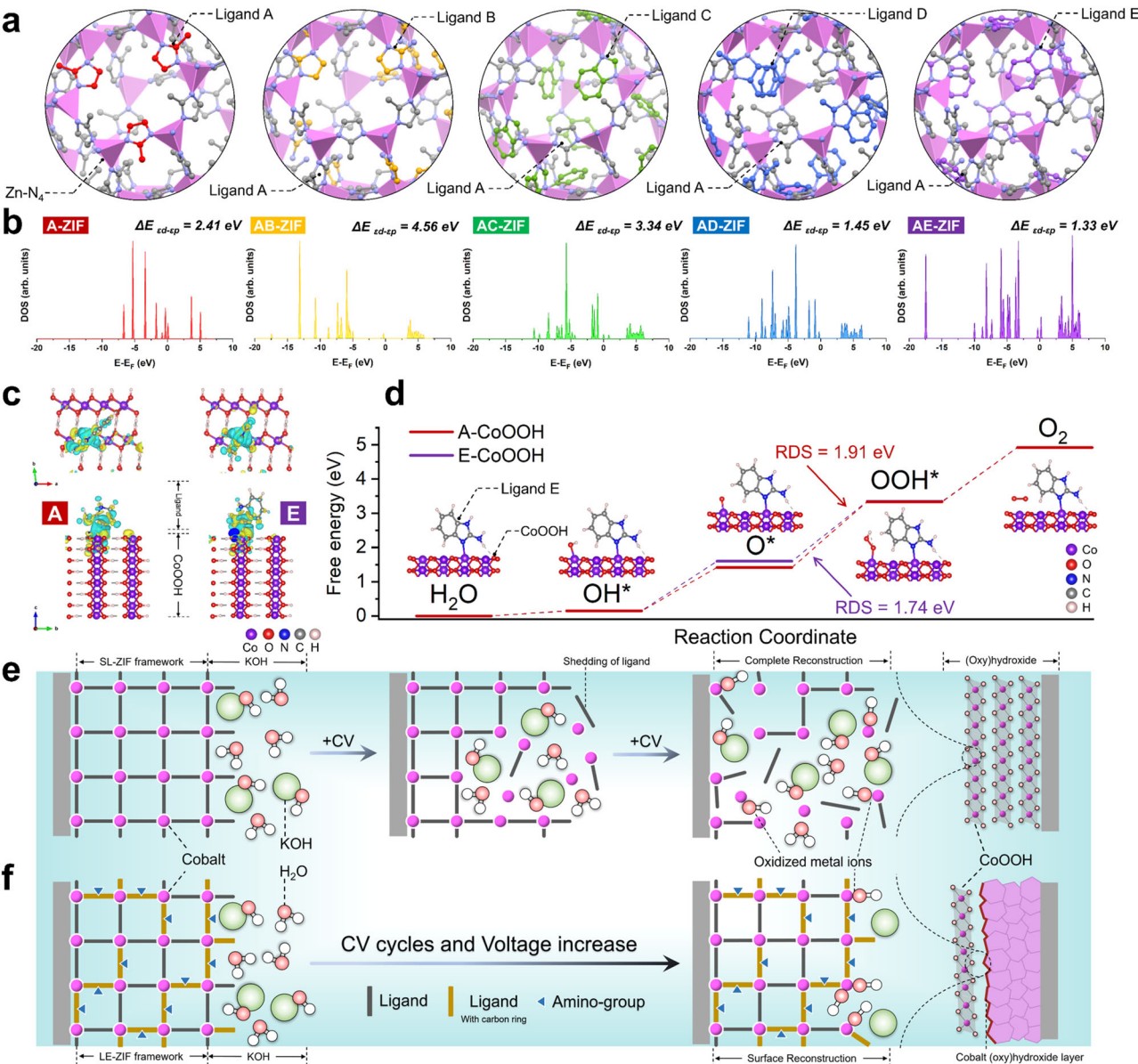

**Fig. 7 | DFT simulations and mechanisms diagram of as-prepared ZIFs reconstruction. a**, **b** Optimized model structures (**a**) and calculated total DOS (**b**) of A-ZIF, AB-ZIF, AC-ZIF, AD-ZIF and AE-ZIF. **c** Charge density difference of the CoOOH incorporating ligands A and E models. **d** Free energies diagram of the CoOOH incorporating ligands A and E models. More details of the structure models and the free energy calculation steps are provided in Figs. S46–54, Supplementary Data 1, and Supplementary Data 2. **e**, **f** Schematic diagram of the complete reconstruction mechanism (**e**) and surface reconstruction mechanism (**f**) in (photo)electrocatalytic reaction.

incorporating ligand E, the energy barrier for the rate-determining step (RDS, O* oxidized to OOH*) of CoOOH model decreases from 2.02 eV to 1.74 eV, lower than ligand A at 1.91 eV (Fig. 7d and Fig. S54). This implies there is likely a synergistic interaction between the ligand E and CoOOH at the reconstructed interface. In summary, the ligand E mixed in AE-ZIF probably play a vital role enhancing OER activity, featuring both additional -NH₂ groups and π-π stacking aromatic carbon rings, which provide superior electronic absorption and transfer capabilities compared to ligands B, C and D.

Figure 7e shows a schematic representation of the CR mechanism as occurring in A and AB. Upon weakening of the ligand coordination, the Co species either leach into the solution or directly react with OH⁻ ions from the alkaline electrolyte to form (oxy)hydroxide compounds. This phase transition process is associated with changes in morphology and surface chemistry that result in a complete loss of porosity as

well as a decrease in the number of available active sites and, consequently, activity towards electrocatalytic OER. Moreover, this conversion leads to an increase in band gap and, thus, a loss of ability to absorb visible light.

On the other hand, the SR mechanism inherent to AC, AD and AE is depicted in Fig. 7f. Here, the framework structure is largely preserved due to the stronger ligand-metal coordination due to the added secondary ligand, as represented by lower potential energies. Therefore, the phase transformation to CoOOH is confined to a thin layer on the particles surface (Fig. 4e). After 2 h, the activity tends to stabilize, indicating the establishment of a stable protective core-shell structure, with the inner layer as the core retaining the inherent framework structure. Outer (oxy)hydroxide layer enables to maintain the electrode durability during electrocatalysis in alkaline environment, enhancing conductivity and promoting electron transport under large current density[49,59].

## Discussion

In summary, we have constructed a series of mixed-ligand ZIF catalysts through ligand engineering to investigate the (photo)electrocatalytic OER mechanisms in alkaline environments. Among these catalysts, AE-ZIF engineered with ligand 2-aminobenzimidazole, showcased electrocatalytic activity that was an order of magnitude higher than the single-ligand A-ZIF, with an 8-fold increase in current density and a 7-fold increase in $O_2$ evolution rate. Similarly, the same behaviour was observed in photoelectrocatalysis, with performance of AE-ZIF under visible light about doubling compared to dark conditions.

Incorporating the secondary ligand, particularly in AE-ZIF, significantly enhances the structural robustness in (photo)electrocatalytic cases, maintaining electrochemical stability for at least 12 h compared to the degradation observed in single-ligand A-ZIF after only 2 h. Advanced electrochemical techniques, including in-situ UV–vis absorption, Raman and Photoluminescence spectroscopy, reveal a complete mechanism induced by the addition of a secondary ligand.

Samples A-ZIF and AB-ZIF undergo complete reconstruction leading to the formation of additional phases, i.e. (oxy)hydroxide nanosheets and an amorphous phase, resulting in significant loss of porosity, electrical conductivity and visible light absorption capabilities. In contrast, mixing a secondary ligand with π-π stacking aromatic carbon rings, i.e AC-ZIF, AD-ZIF and AE-ZIF, largely preserved the inherent framework structure and porosity, forming a thin stable (oxy)hydroxide layer on the particle surface. This outer layer effectively protected and passivated the ZIF framework, by creating a more stable core-shell structure.

Theoretical calculations further underscored the synergistic effect between the secondary ligand and the (oxy)hydroxide layer, which likely accelerates electron transfer and reduce Gibbs free energy, ultimately enhancing overall OER activity. Additionally, the unique functional groups (aromatic carbon ring and amino group) in the secondary ligands can improve the electronic conductivity and strengthened the Co-N covalency. This mixed-ligand engineering has achieved notable enhancements and offers a feasible pathway for designing more active and stable ZIFs, extending applications beyond electro- and photoelectrocatalysis.

## Methods

### Chemicals

Cobalt nitrate hexahydrate ($Co(NO_3)_2 \cdot 6H_2O$, 99%, STREM), 2-methylimidazole (2-mIm, $C_4H_6N_2$, 99%, Sigma-Aldrich), 1H-Imidazol-2-amine ($NH_2$-mIm, $C_3H_5N_3$, 97%, BLD pharm), Benzimidazole (bIm, $C_7H_6N_2$, 98%, Sigma-Aldrich), 2-methylbenzimidazole (2-bIm, $C_8H_8N_2$, 98%, abcr GmbH), 2-aminobenzimidazole ($NH_2$-bIm, $C_7H_7N_3$, 97%, Sigma-Aldrich), Methanol (MeOH, 99.9%, HiPerSolv CHROMANORM, VWR), Postassium hydroxide (KOH, 85%, Sigma-Aldrich), Nafion™ perfluorinated resin solution (nafion, 5 wt.%, Sigma-Aldrich), $d_4$-acetic acid for NMR ($CD_3CO_2D$, 99.4%, Thermo scientific).

### A-ZIF

2 g of 2-mIm was dissolved in 10 ml of HPLC methanol, while 0.87 g of $Co(NO_3)_2 \cdot 6H_2O$ was dissolved in 10 ml of HPLC methanol. The two solutions were then mixed in microwave synthesis reactor (Monowave 300, Anton Paar), at 150 °C for 2 h to accelerate the growth of A-ZIF crystals. After being allowed to deposit for 24 h, the powder was centrifuged, washed three times with methanol and DI water, and then vacuum dried in an oven overnight, resulting in a purple A-ZIF powder.

### AB-ZIF

1 g of $NH_2$-mIm and 1 g of 2-mIm were mixed into 10 ml of HPLC-grade methanol. The solution was placed into a microwave synthesis reactor (Monowave 300, Anton Paar) and heated to 100 °C for 30 min to ensure even mixing and dissolution of the ligands. Meanwhile, 0.87 g of $Co(NO_3)_2 \cdot 6H_2O$ was dissolved in 10 ml of HPLC-grade methanol,

sonicated for 10 min, and stirred for 30 min. The two solutions were then combined and heated in the microwave synthesis reactor at 150 °C for 2 h to accelerate the growth of ZIF crystals. After 24 h of deposition, the resulting powder was centrifuged and washed three times with methanol and deionized water. Finally, AB-ZIF was obtained by vacuum drying overnight.

### AC-ZIF

1 g of bIm and 1 g of 2-mIm were mixed into 10 ml of HPLC-grade methanol. The solution was placed into a microwave synthesis reactor (Monowave 300, Anton Paar) and heated to 100 °C for 30 min to ensure even mixing and dissolution of the ligands. Meanwhile, 0.87 g of $Co(NO_3)_2 \cdot 6H_2O$ was dissolved in 10 ml of HPLC-grade methanol, sonicated for 10 min, and stirred for 30 min. The two solutions were then combined and heated in the microwave synthesis reactor at 150 °C for 2 h to accelerate the growth of ZIF crystals. After 24 h of deposition, the resulting powder was centrifuged and washed three times with methanol and deionized water. Finally, AC-ZIF was obtained by vacuum drying overnight.

### AD-ZIF

1 g of 2-bIm and 1 g of 2-mIm were mixed into 10 ml of HPLC-grade methanol. The solution was placed into a microwave synthesis reactor (Monowave 300, Anton Paar) and heated to 100 °C for 30 min to ensure even mixing and dissolution of the ligands. Meanwhile, 0.87 g of $Co(NO_3)_2 \cdot 6H_2O$ was dissolved in 10 ml of HPLC-grade methanol, sonicated for 10 min, and stirred for 30 min. The two solutions were then combined and heated in the microwave synthesis reactor at 150 °C for 2 h to accelerate the growth of ZIF crystals. After 24 h of deposition, the resulting powder was centrifuged and washed three times with methanol and deionized water. Finally, AD-ZIF was obtained by vacuum drying overnight.

### AE-ZIF

1 g of $NH_2$-bIm and 1 g of 2-mIm were mixed into 10 ml of HPLC-grade methanol. The solution was placed into a microwave synthesis reactor (Monowave 300, Anton Paar) and heated to 100 °C for 30 min to ensure even mixing and dissolution of the ligands. Meanwhile, 0.87 g of $Co(NO_3)_2 \cdot 6H_2O$ was dissolved in 10 ml of HPLC-grade methanol, sonicated for 10 min, and stirred for 30 min. The two solutions were then combined and heated in the microwave synthesis reactor at 150 °C for 2 h to accelerate the growth of ZIF crystals. After 24 h of deposition, the resulting powder was centrifuged and washed three times with methanol and deionized water. Finally, AE-ZIF was obtained by vacuum drying overnight.

### ZIF@FTO glass electrode

The ZIF powders are prepared by drop-casting onto the fluorine-doped tin oxide (FTO) glass surface. To prepare the mixture, 3 mg of the ZIF samples are combined with 280 μl of HPLC methanol and 20 μl of nafion solution, and the mixture is sonicated for 30 min. Subsequently, the mixed solution is dropped onto the surface of the three FTO glass (1 cm × cm). This entire process is repeated to obtain three ZIF@FTO electrodes. Finally, the ZIF@FTO electrodes are placed in a vacuum oven and dried overnight. The same procedure is applied to prepare electrodes for each ZIF@FTO sample.

### Material characterizations

X-ray diffraction (XRD) were carried out using a PANalytical X'Pert Pro multi-purpose diffractometer (MPD) with Bragg Brentano geometry, equipped with a Cu anode at 45 kV and 40 mA, a BBHD Mirror, and an X-Celerator multichannel detector[60]. X-ray photoelectron spectroscopy were carried out using a custom-built SPECS XPS spectrometer, featuring a monochromatized Al-Kα X-ray source (μFocus 350) and a hemispherical WAL-150 analyzer (acceptance angle: 60°)[61]. The high-

resolution transmission electron microscopy (TEM) images presented in this paper were acquired on a Tecnai F20 FEG-TEM at USTEM (university service center for transmission electron microscopy) at TU Wien, equipped with an X-FEG, a Gatan Rio16 CCD-camera, a Gatan DigiSTEM II with HAADF detector for STEM imaging, and an EDAX-AMETEK Apollo XLTW SDD EDX-detector. Liquid phase Nuclear magnetic resonance spectroscopy (NMR) was conducted using a Bruker ADVANCE 250 (250.13 MHz) instrument, equipped with a 5 mm inverse-broad probe head and z-gradient unit[29].

In-situ electrochemical Raman spectroscopy was performed with a HORIBA LabRAM spectrometer with a 532 nm laser, in-situ electrochemical UV–Vis absorption spectroscopy was obtained at 350–700 nm by Jasco V-670, and in-situ electrochemical Photoluminescence spectroscopy was carried out with a PicoQuant FluoTime 300 spectrophotometer[60]. All in-situ measurement were conducted within a three-electrode configuration, with the working electrode securely positioned in a specialized electrolytic cell. For more characterization details as described in the Supplementary Section S8.

## (Photo)electrochemical measurements

Potentiostats (Autolab AUT85726 and CHI 760E) are used to perform all electrochemical experiments in a three-electrode configuration. Electrocatalytic cell used in the experiments was a 231 mL sourced from PerfectLight, Ltd. A saturated Ag/AgCl serving as the reference electrode, and a Pt electrode serves as the counter electrode; both are sourced from Gaoss Union, Ltd. 1 M KOH solution is used as the electrolyte (pH 13.721 ± 0.01), prepared by dissolving an appropriate amount of potassium hydroxide in deionized water, then saturated it with nitrogen gas and stored it at room temperature. The working electrode consists of ZIF@FTO glass, including all in-situ electrochemical measurements. The potential is converted to the reversible hydrogen electrode (RHE) using the Nernst equation: $E_{RHE} = E_{Ag/Ag/Cl} + 0.21 + 0.059 \times pH$. Cyclic voltammetry (CV) cycles are tested at a scan rate of 10 mV s$^{-1}$. Three potential windows are selected: a low potential window of 0.85–1.05 V, a medium potential window of 0.85–1.30 V, and a high potential window of 0.85–1.55 V (vs. RHE). Electrochemical impedance spectroscopy (EIS) is measured with a frequency range from $10^5$ – 0.1 Hz. All data are not calibrated with the iR correction.

Conversion formula for the current density unit of each samples from A g$^{-1}$ to mA cm$^{-2}$ is: 3.33 A g$^{-1}$ = 10 mA cm$^{-2}$. Using this formula, the overpotential for AE-ZIF at 10 mA cm$^{-2}$ is 0.3 V, while A-ZIF only reaches 2 mA cm$^{-2}$ at the same potential.

In-situ $O_2$ flow detection configuration was employed an argon gas to continuously purge the cell reactor (same cell as electrocatalysis) at a flow rate of 30 mL min$^{-1}$, delivering $O_2$ to the X-stream analyzer (Emerson Process Management). The $O_2$ concentrations (ppm) measured in the stream is converted to the $O_2$ evolution rate (μmol h$^{-1}$) by the ideal gas equation.

Photoelectrocatalytic experiments were performed using visible light (400–700 nm, 2.36 W) from SUPERLITE SUV-DC-E (Lumatec) as a visible light source.

Electrochemical surface area (ECSA) is directly proportional to its electrochemical double-layer capacitance according to the following equation: ECSA = $C_{dl}/C_s$, where $C_s$ is the specific capacitance of the sample, 1 M KOH is usually taken with $C_s$ assumed to be 0.04 mF cm$^{-2}$[62]. $C_{dl}$ is the electrochemical double-layer capacitance of the sample, estimated according to the above CV curves of the non-Faraday region[63].

Electrical resistance of each ZIF@FTO samples were estimated using a two-probe measurement method with the probes positioned primarily on the FTO glass. The conversion equation for electrical conductivity ($\sigma$) and resistance ($R$) is $\sigma = l/RA$, where $l$ is the thickness of the ZIFs, and $A$ is the area covered by the ZIFs[64].

Reference electrode preparation and calibration, first the silver wire was ground using 240-grit SiC sandpaper, then rinsed with deionized water, and dipped into a 1 M HCl solution. The electrochemical deposition of AgCl was then conducted with the following setup: CV test with a scan from 0.7 to 0.2 V (vs. Ag/AgCl/3 M KCl) for 10 cycles, followed by amperometry at 0.3 V for 2 min, and another test at 0.7 V for 15 min.

## Calculation of band gap, LUMO, and HUMO

Band gap was calculated with the equation proposed by Tauc/Davis and Mott et al.:

$$(\alpha h \upsilon)^{1/n} = A(h \upsilon - E_g)$$

Where $\alpha$ is absorbance index, $h$ is Planck's constant, $\upsilon$ is the frequency, $A$ is a constant, $E_g$ is the semiconductor band gap, and $n$ is related to the related to band gap type ($n = 1/2$ for direct band gap and $n = 2$ for indirect band gap)[32].

Flat band potential was calculated with the following equation:

$$\frac{1}{C_{sc}^2} = 2\left(\frac{\triangle\Phi_{sc} - RT/F}{q\varepsilon\varepsilon_0 N}\right)$$

Where $C_{sc}$ is the charge capacitance, $\Delta\Phi_{sc}$ is the absolute value of difference between the electrode potential and the flat band potential, $\varepsilon$ is the relative dielectric constant, $\varepsilon_O$ is the vacuum dielectric constant, $N$ is the donor (for n-type semiconductor) or acceptor (for p-type semiconductor) density[32]. The flat band potential, determined as the point where the tangent of the Mott–Schottky plot intersects the horizontal axis, is about equal to the position of the lowest unoccupied molecular orbital (LUMO) potential for ZIFs, which behave similarly to n-type semiconductors[65]. Highest occupied molecular orbital (HOMO) potential can be then calculated based on the known LUMO potential and their band gap.

## Density functional theory (DFT) calculations

All calculations were conducted using Vienna ab initio program package (VASP)[66]. Electron exchange and correlation effects were described using the Perdew-Burke-Ernzerhof (PBE) functional[67], which is part of the generalized gradient approximation (GGA) method[68]. The projector augmented wave (PAW) approach was employed to describe the interaction between electrons and ions, with spin polarization taken into account. A plane-wave cutoff energy of 500 eV was applied. A ($3 \times 3 \times 1$) $k$-point mesh was used for k-space integration in ligand binding to cobalt hydroxide structure relaxations. For the structure relaxation of MOF structures with different ligands, a ($1 \times 1 \times 1$) $k$-point mesh was used, while a ($4 \times 4 \times 4$) $k$-point mesh was applied for DOS calculation. The conjugate-gradient algorithm was used to relax the ions into their instantaneous ground state. All structures were fully relaxed with energy and force convergences of $<1 \times 10^{-6}$ eV and 0.03 eV Å$^{-1}$, respectively.

For the calculation of OER free energy diagram, a vacuum layer of 20 Å was set to mitigate the influence of crystal periodicity. The widely accepted method to model the OER process involves a four-electron reaction pathway:

$$H_2O + ^* \rightarrow OH^* + H^+ + e^-$$

$$OH^* \rightarrow O^* + H^+ + e^-$$

$$H_2O + O^* \rightarrow OOH^* + H^+ + e^-$$

$$OOH^* \rightarrow ^* + O_2 + H^+ + e^-$$

where * represents an active site on the bare catalyst surface, while OH*, O*, and OOH* denote different intermediates[69].

For each elementary step, the Gibbs free energy $\Delta G_i$ (i = 1, 2, 3, 4) can be calculated as:

$$\Delta G_i = \Delta E + \Delta ZPE - T\Delta S$$

where $\Delta E$ is the total energy difference between the reactant and product molecules in the four steps, $\Delta ZPE$ is the change in zero-point energy, and $T\Delta S$ is the contribution of entropy[70].

## Data availability

All data supporting the findings of this study are available within the article and its Supplementary Information file, as well as available from the corresponding author upon reasonable request. Source data are provided with this paper.

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

## Acknowledgements

Z.H. was financially supported by China Scholarship Council (Nos. 202106770017). This research was supported by Austrian Science Fund (FWF: I5413-N, Cluster of Excellence MECS: 10.55776/COE5, and doc-toral college TU-DX: 10.55776/DOC142) and Natural Science Foundation of China (Nos. U20A20246 and 52472205). The authors gratefully acknowledge the services provided by the analytical instrumentation center (AIC) at TU Wien, X-ray Center (XRC) at TU Wien, University Ser-vice for Transmission Electron Microscopy at TU Wien (USTEM).

## Author contributions

Z.H. contributed most of the experiments and analysis and wrote the initial manuscript. Z.W. contributed in-situ electrochemical Raman

analysis, reviewing & editing the manuscript. H.R. contributed XPS measurement and analysis, reviewing & editing the manuscript. S.N. contributed $N_2$ physisorption and $^1$H NMR measurement, reviewing & editing the manuscript. Q.Z. contributed DFT simulations. S.S. contributed TEM measurement. D.H.A. contributed reviewing & editing the manuscript. Y.Y. contributed experimental equipment, reviewing & editing the manuscript. D.E. contributed the project concept, funding and experimental equipment and reviewing & editing the manuscript. All authors discussed the results and commented on the manuscript.

## Competing interests

The authors declare no competing interests.
