## [Peer Review file · Nature Communications]

Ligand Engineering Enhances (Photo)Electrocatalytic Activity and Stability of Zeolitic Imidazolate Frameworks via In-situ Surface Reconstruction

Corresponding Author: Professor Dominik Eder

Version 0:

Reviewer comments:

Reviewer #1

(Remarks to the Author)

This work reported a ligand engineering strategy to enhance the (photo)electrocatalytic activity of ZIF toward oxygen evolution reaction (OER). By introducing mixed ligands, the ZIF framework structure can be preserved to exhibit stable and efficient performance toward OER. The topic of the work is significant with regard to the development of MOF-based electrocatalysts. The manuscript can be considered for publication after the following major comments are addressed.

- (1) For most of the OER catalysts, the surface reconstruction is an inevitable process that ensures the stable and efficient electrocatalytic performance. Without this process, the catalysts, even in the reconstructed form, are subjected to inactivation during the OER process. For the amperometry tests, would AE-ZIF still show stable current density and oxygen yield under intermittent biased condition, i.e. applying 1.65 V for 3 h, followed by ceasing the bias and subsequently applying 1.65 V for another 3 h?
- (2) For amperometry data, why was the unit of current density of A per g used? The unit of A per cm² represents an index for global comparison.
- (3) Did the cobalt (oxy)hydroxide layer fully cover the surface of AE-ZIF? If not, the protective effect might be minor.
- (4) A table summarizing the global performance comparison, in particular the long-term stability, with the state-of-the-art MOF based electrocatalysts ever reported should be provided. With this comparison, the merits of the current work can be highlighted.
- (5) In Fig S14, the Nyquist plots must be presented as square with equal scale at x-axis (Z') and y-axis ($-Z''$). Otherwise, the plots would be distorted to cause misinterpretations on the capacitance behaviors

Reviewer #2

(Remarks to the Author)

In this manuscript, Huang et al. synthesized four new mixed-ligand ZIF catalysts through ligand engineering, and compared them with the single-ligand A-ZIF. Among these, the ZIF designed with new ligand 2-aminobenzimidazole showcased great OER activity and stability in electrocatalysis. The authors used in-situ Raman and TEM to demonstrate that this is due to the formation of an ultrathin stable (oxy)hydroxide layer via in-situ reconstruction. Additionally, the photoelectrocatalytic properties of each ZIF samples were investigated using in-situ PL and UV-vis spectroscopy. Finally, the authors emphasized the critical role of unique functional groups in the new ligands through DFT calculations, and elucidated the reconstruction mechanisms induced by different ligands in electrocatalysis. Overall, I this work is interesting and is supported by extensive experimental data, providing valuable insights for future MOF design and ligand selection. Thus, I recommend it publication pending appropriate responses to the following comments.

1. For MOF materials, the changes in porosity and active surface area during reactions are crucial. The authors have only provided physical adsorption data before the reaction, and those after the reaction are needed for comparison purposes.
2. Post-reaction IR spectroscopy data or in-situ IR analysis to investigation the reconstruction process of ZIF should be provided.
3. Single-ligand A-ZIF exhibited an interesting trend of OER activity in electrocatalysis, initially increasing and then

decreasing. What are the detailed reasons for this outcome? Is it solely caused by phase transition, or are there other factors at play?

4. An in-situ stable (oxy)hydroxide layer was formed for AE-ZIF during the surface reconstruction process. Does the thickness of this layer affect the performance of your samples? Can you calculate the thickness of the layer formed at different potentials?

5. There are Co(OH)₂ and CoOOH in the (oxy)hydroxide layer for AE-ZIF and AD-ZIF, which were not prominently observed in XRD and XPS but were detected in Raman spectroscopy. More detailed reasons for this should be discussed.

6. Figure 1e shows the ¹H NMR spectrum of ZIFs. The main text should specify the solvent used for digestion, d₄-acetic acid?

7. In Figure 3 of the XPS data, it is necessary to clearly label which peaks represent Co-N, C-N and other relevant bonds. Similarly for N 1s. Specifically, indicate with distinct colors or labels: blue for Co-N and red for C-N. Additionally, the potential window is too small to read, so the font needs to be adjusted up.

8. In the schematic diagrams of DFT calculations shown in Figures 5c and 5d, it is necessary to directly annotate the colors representing different atoms. Additionally, they are too close to other figures; it is recommended to separate them to avoid confusion.

9. The authors are recommended to read the following papers (Appl. Catal. B-Environ. Energy 2024. 350: 123925; Chem. Eng. J. 2023, 455: 140601) for improving the manuscript.

Reviewer #3

(Remarks to the Author)

Authors synthesized four distinct mixed-ligand versions of zeolitic imidazolate framework-67 (ZIF-67) and conducted a comprehensive investigation into the structural evolution and degradation processes during electrocatalytic oxygen evolution reactions. The key point, surface reconstruction to form a protective cobalt (oxy)hydroxide layer, boosts both OER activity and stability. This scope is interesting and fundamental, and the whole story sounds good. This work is well organized and the results are important and convincing. Therefore, I would like to recommend its publication on Nature Communications after following minor questions being responded.

1. Both AC and AD samples show enhanced stability, however, their activity is lower than AE. What is the reason for this difference? Is their performance determined by the initially mixed new ligands?

2. In Figure 2, the O₂ evolution curves show a significant upward trend within the initial time. Is this caused by reconstruction or the accumulation of O₂ in the device?

3. The characterization, which is conducted to explore the reconstruction process, shows a decreased signal intensity for the characteristic peaks. What is the reason? Is there structural degradation?

4. In Figure 1, the authors should show the full IR spectrum and note which signals originate from the ligands and which are from the metal-ligand bonding.

5. A line scan analysis is suggested to conduct to show the core-shell structure of AE-ZIF in Figure 4.

6. Many of the annotations in Figure 5 are too small to read.

7. The overpotential of catalyst at 10 mA cm⁻² is an important reference to evaluate the activity. The authors could present this basic data.

8. How do the authors construct the ZIF models with mixed ligands in DFT calculations? It is suggested to provide the input files for each ZIF structure considered in this manuscript.

9. The following reports are for the references to improve the manuscript: Appl. Catal. B-Environ. 2024 , 353 , 124096; J. Colloid Interface Sci. 2024, 653: 380; Angew. Chem. In. Ed. 2022, 134: e202214794.

Version 1:

Reviewer comments:

Reviewer #1

(Remarks to the Author)

The revised manuscript is now in a good shape for publication.

Reviewer #2

(Remarks to the Author)

This manuscript has been carefully revised according to reviewers' comments. It should be accepted to published on Nature Communications.

Reviewer #3

(Remarks to the Author)

The manuscript was revised according to the suggestions very well and I considered that the revised manuscript could be accepted to be published in Nature Communications

Point-by-point response to Reviewers' comments

Reviewer #1

General comment. This work reported a ligand engineering strategy to enhance the (photo)electrocatalytic activity of ZIF toward oxygen evolution reaction (OER). By introducing mixed ligands, the ZIF framework structure can be preserved to exhibit stable and efficient performance toward OER. The topic of the work is significant with regard to the development of MOF-based electrocatalysts. The manuscript can be considered for publication after the following major comments are addressed.

Response: We thank the reviewer for the positive assessment. We have answered all the questions and doubts you expressed below.

1. For most of the OER catalysts, the surface reconstruction is an inevitable process that ensures the stable and efficient electrocatalytic performance. Without this process, the catalysts, even in the reconstructed form, are subjected to inactivation during the OER process. For the amperometry tests, would AE-ZIF still show stable current density and oxygen yield under intermittent biased condition, i.e. applying 1.65 V for 3 h, followed by ceasing the bias and subsequently applying 1.65 V for another 3 h?

Response: We fully agree that surface reconstruction is a crucial step to ensure the stable and efficient electrocatalytic performance of ZIF. It is thus a good idea to test the current density and O₂ evolution rate of AE-ZIF under intermittent biased conditions using amperometry.

Therefore, we conducted additional amperometry experiments with AE-ZIF: we first applied a potential of 1.65 V for 3 hours, then the bias was ceased for 3 hours, and finally, 1.65 V was reapplied for another 3 hours (**Fig. S13d**). Initially, the trends in current density and O₂ evolution rate during the first 3 hours were consistent with the descriptions in our manuscript. The slow increase in current density was attributed to surface reconstruction forming a high-valence cobalt (oxy)hydroxide layer, while the rise in oxygen evolution rate resulted from the continuous accumulation of O₂ in the reaction cell.

When the potential was ceased, the current density quickly dropped to zero, and the oxygen evolution rate gradually declined. Importantly, upon reapplying 1.65 V, the current density rapidly returned to its original level and remained stable. Unlike the initial 3 hours, AE-ZIF did not undergo a slow increase in current density due to surface reconstruction again,

indicating that AE-ZIF had already developed a stable cobalt (oxy)hydroxide layer, which inhibited further electro-oxidation, achieving a structurally stable state. Similarly, the gradual rise in the O₂ evolution rate was due to the O₂ accumulation in the cell.

Based on the results of the intermittent biased amperometry test, we can now specify the process further and conclude that AE-ZIF completes surface reconstruction within the first two hours, forming a stable cobalt (oxy)hydroxide layer that effectively inhibits further electro-oxidation process, subsequently maintaining stable current density and O₂ evolution rate.

Page 8: In contrast, the structural reconstruction in AE-ZIF ceased after 2 hours and maintained a more stable current under constant potential or intermittent bias, exhibiting a current density and O₂ evolution rate that were an order of magnitude higher than those of A-ZIF (Figs. 2e and S13d).

Supporting Info Page 12: Under intermittent biased conditions, after stopping and restarting the application of 1.65 V, the current density and O₂ evolution rate of AE-ZIF quickly returned to their original levels and remained stable (Fig. S13d). Unlike the initial three hours, AE-ZIF did not exhibit a gradual increase in current density due to surface reconstruction again, indicating that AE-ZIF formed a stable cobalt (oxy)hydroxide layer within the first two hours. This layer effectively inhibited further electro-oxidation process, achieving in a structurally stable state.

Supporting Info Page 20:

Fig. S13. d, Amperometric and O₂ evolution rate plots of AE-ZIF under intermittent biased condition.

2. For amperometry data, why was the unit of current density of A per g used? The unit of A per cm² represents an index for global comparison.

Response: We agree with your point that using the current density unit of mA/cm² is generally the standard for global comparison of electrocatalysts. However, we have chosen to use A/g for our manuscript based on the following considerations:

1. We used a drop-casting method to load the ZIF powder onto FTO glass as the electrode. This allows us to precisely know the amount of powder used, i.e., the specific mass of each ZIF sample on the conductive FTO area. Due to the nature of drop-casting, as opposed to spin-coating, it is difficult to achieve a uniform distribution of ZIF powder over the large area (3 cm × 1 cm) of FTO glass. Therefore, we consider that using the loading ZIF mass on FTO as the denominator is more accurate than using the geometric area of FTO. We avoided the spin-coating method to prevent significant sample loss and to control the consistent ZIF amount on each FTO electrode, which beneficial for our subsequent studies of the reconstruction mechanism.
2. Using the geometric area of FTO or other large-area electrodes as the denominator may not accurately reflect the intrinsic activity of the electrocatalyst ¹. If a small-contact-area glassy carbon electrode (GCE) were used, the unit of mA/cm² would more accurately reflect the real electrocatalytic activity. However, with a fixed geometric area of the FTO electrode, the current density of the electrocatalyst can be significantly affected by the loading mass. FTO electrodes facilitate our photoelectrocatalysis and in-situ spectroscopy tests, and we prefer to maintain the same type of electrode throughout all experiments.
3. The adoption of A/g allows our characterization results across various MOF-based electrocatalytic studies, especially for cyclic voltammetry tests and subsequent in-situ electrochemical analyses. Currently, report ² on MOFs mechanism also are increasingly favoring the use A/g rather than mA/cm².

Based on these reasons, we chose to use A/g for current density rather than mA/cm². However, we acknowledge your point that mA/cm² is a globally recognized metric that is widely accepted by readers. Therefore, to facilitate better understanding and comparison of our catalyst's performance, we have provided the conversion formula of current density from A/g to mA/cm².

Supporting Info Page 73:

Conversion formula for the current density unit of our samples from A g^{-1} to mA cm^{-2} is: $3.33 \text{ A g}^{-1} = 10 \text{ mA cm}^{-2}$. Using this formula, the overpotential for AE-ZIF at 10 mA cm^{-2} is 1.53 V , while A-ZIF only reaches 2 mA cm^{-2} at the same potential.

3. Did the cobalt (oxy)hydroxide layer fully cover the surface of AE-ZIF? If not, the protective effect might be minor.

Response: This is an excellent and important comment. Based on a series of experiment evaluations, we consider that the most AE-ZIF particles' surfaces are fully covered by (oxy)hydroxide layer, forming a complete core-shell structure. The ZIF structure is sensitive to electron beam irradiation by TEM, which often leads to some losses likely. This is why, in **Figs. 4d** and **4f**, we only showed AE-ZIF particles that were not completely covered by cobalt (oxy)hydroxide. It is crucial to emphasize that the cobalt (oxy)hydroxide layer described in this manuscript was in-situ grown during electrochemical OER environment. If specific edges of AE-ZIF particles remain in contact with the electrolyte through applied potential, the electro-oxidation will likely continue until the layer fully covers the core ZIF structure..

Fig. 4: **d**, TEM image of single-particle AE-ZIF after 12 hour-amperometry. **e**, Schematic diagrams and enlarged TEM images of AE-ZIF edge after 12 hour-amperometry, the lower corresponding FFT patterns. **f**, Elemental mapping of single-particle AE-ZIF after (f) 12 hour-amperometry showing the Co, O, N and C distribution.

4. A table summarizing the global performance comparison, in particular the long-term stability, with the state-of-the-art MOF based electrocatalysts ever reported should be provided. With this comparison, the merits of the current work can be highlighted.

Response: Thank you, we fully agree that comprehensive studies regarding the long-term electrocatalytic stability of our ZIFs would greatly increase the merits and impact of our manuscript. A comparative study with other MOF-based electrocatalysts in the field, as requested, however, would be a substantial effort that requires to change the electrode from FTO to either GCE or NF and to redo all measurements, which is out-of-scope for this work. In addition, changing the electrode does not ensure good consistency between electrocatalytic testing and in-situ electrochemical analysis.

In this work, we focused on the effects of ligand engineering on reconstruction and consequently the (photo)electrocatalytic performance, involving conductivity, OER activity and stability. For practical and safety reasons, we stopped the measurements after only 12 hours. However, we agree that the performance of AE-ZIF is likely stable for considerably longer durations than 12 hours. Under the same experimental conditions, the surface-reconstructed AE-ZIF exhibited an order of magnitude higher OER activity and better stability compared to single-ligand A-ZIF, which has provided strong evidence for the advantages of the ligand engineering strategy in OER electrocatalysis.

5. In Fig S14, the Nyquist plots must be presented as square with equal scale at x-axis (Z') and y-axis ($-Z''$). Otherwise, the plots would be distorted to cause misinterpretations on the capacitance behaviors.

Response: Thank you very much for your comment, and I apologize for not realizing the issue with the Nyquist plots earlier. I have now modified the Nyquist plots to be square and adjusted the scales of the x-axis (Z') and y-axis ($-Z''$) to be equal.

Supporting Info Page 22:

Fig. S15. Electrochemical impedance spectroscopy of A-ZIF (a), AB-ZIF (b), AC-ZIF (c), AD-ZIF (d) and AE-ZIF (e) on FTO glass between 0.85-1.55 V with different CV cycles (0, 10, 20, 30, 40, 50, 60, 70, 80, 90 and 100).

Reviewer #2

General comment. In this manuscript, Huang et al. synthesized four new mixed-ligand ZIF catalysts through ligand engineering, and compared them with the single-ligand A-ZIF. Among these, the ZIF designed with new ligand 2-aminobenzimidazole showcased great OER activity and stability in electrocatalysis. The authors used in-situ Raman and TEM to demonstrate that this is due to the formation of an ultrathin stable (oxy)hydroxide layer via in-situ reconstruction. Additionally, the photoelectrocatalytic properties of each ZIF samples were investigated using in-situ PL and UV-vis spectroscopy. Finally, the authors emphasized the critical role of unique functional groups in the new ligands through DFT calculations, and elucidated the reconstruction mechanisms induced by different ligands in electrocatalysis. Overall, I this work is interesting and is supported by extensive experimental data, providing valuable insights for future MOF design and ligand selection. Thus, I recommend it publication pending appropriate responses to the following comments.

Response: We thank the reviewer for the positive comment. Your questions have been very insightful to us and allowed us to further improve the structure of the manuscript.

1. For MOF materials, the changes in porosity and active surface area during reactions are crucial. The authors have only provided physical adsorption data before the reaction, and those after the reaction are needed for comparison purposes.

Response: We agree with your suggestion that comparing the physical adsorption data before and after the reaction would reveal the effect of electrocatalysis on the ZIF porosity. However, considering that the amount of ZIF sample on the electrode after electrocatalytic reaction is usually insufficient for a complete physical adsorption measurement, providing post-reaction physical adsorption data is quite difficult. Nevertheless, we can infer that the inherent porosity and specific surface area of the ZIF samples probably decrease due to the disruption of the framework structure caused by electro-oxidation. This is further supported by the complete reconstruction of A-ZIF into cobalt (oxy)hydroxide nanosheets, indicating a complete disruption of the ZIF porous structure.

To address this, we have included additional experiments to follow changes in the electrochemical active surface area (ECSA). Although the principle of ECSA differs significantly from that of specific surface area obtained from physical adsorption, it still provides a substantial evaluation of the catalyst's surface area during electrocatalysis¹. Our method for assessing the ECSA of the samples is directly proportional to the

electrochemical double-layer capacitance (C_{dl}), which was estimated based on the non-Faradaic region of the cyclic voltammetry (CV) curves at different scan rates (20 to 100 mV s^{-1} ; see **Fig. S9a-e**). As shown in **Figs. S9c** and **S9f**, before the electrocatalytic activation, A-ZIF and AE-ZIF exhibited similar ECSA values of 64.25 cm^{-2} and 78.00 cm^{-2} , respectively. However, after 100 cycles, the ECSA of A-ZIF decreased significantly to 33.25 cm^{-2} due to its own complete reconstruction, representing a reduction of about 48%. In contrast, AE-ZIF, which experienced only surface reconstruction, maintained a high ECSA value of 71.50 cm^{-2} , showing a minor decrease of 8%. These conclusions are consistent with our previous experiments and inferences.

Page 7: When using electrochemical double-layer capacitance (C_{dl}) to reflect the electrochemical active surface area (ECSA, in **Fig. S9**), it is evident that after 100 CV cycles, the reduction in ECSA for AE-ZIF (8%) is less pronounced compared to the significantly decrease observed for A-ZIF (48%). This also indicates that AE-ZIF exhibits stronger resistance properties to continuous electro-oxidation.

Supporting Info Page 11: Since the electrochemically active surface area (ECSA) is directly proportional to the electrochemical double-layer capacitance (C_{dl}), which was estimated based on the non-Faradaic region of the cyclic voltammetry (CV) curves at different scan rates (20 to 100 mV s^{-1} ; see **Fig. S9a-e**). As shown in **Figs. S9c** and **S9f**, before the electrocatalytic activation, A-ZIF and AE-ZIF exhibited similar ECSA values of 64.25 cm^{-2} and 78.00 cm^{-2} , respectively. However, after 100 cycles, the ECSA of A-ZIF decreased significantly to 33.25 cm^{-2} due to its own complete reconstruction, representing a reduction of about 48%. In contrast, AE-ZIF, which experienced only surface reconstruction, maintained a high ECSA value of 71.50 cm^{-2} , showing a minor decrease of 8%. These conclusions are consistent with our partial and complete reconstruction of the two samples, respectively. These conclusions are consistent with our TEM analysis of the two samples. The degradation of the surface area of the AE-ZIF particles caused by the 5 nm thick cobalt (oxy)hydroxide layer (**Fig. 4**) corresponds exactly to a decrease of 8% ECSA, while the decrease of 48% ECSA for A-ZIF corresponding to the transformation of particles into nanosheets.

Supporting Info Page 16:

Fig. S9. Cyclic voltammetry curves of A-ZIF (a), A-ZIF after 100 CV (b), AE-ZIF (d) and AE-ZIF after 100 CV (e) at different scan rates in 1.0 M KOH. Capacitive currents of A-ZIF (c) and AE-ZIF (f). ECSA is directly proportional to its electrochemical double-layer capacitance (C_{dl}) according to the following equation:

$$ECSA = \frac{C_{dl}}{C_s}$$

where C_s is the specific capacitance of the sample, 1 M KOH is usually taken with C_s assumed to be 0.04 mF cm^{-2} ³. where C_{dl} is the electrochemical double-layer capacitance of the sample, estimated according to the above CV curves of the non-Faraday region⁴.

2. Post-reaction IR spectroscopy data or in-situ IR analysis to investigation the reconstruction process of ZIF should be provided.

Response: Thank you for your valuable suggestion. We have now provided new IR spectroscopy data of the post-reaction ZIF, along with corresponding descriptions in the main text, to better elucidate the reconstruction process of the ZIF.

Page 10: This is also evident in the ATR-IR spectra taken after the reconstruction process, which still shows the stretching vibrations from Co-N and 2-mIm ligands in AE-ZIF even after 100 CV cycles, while they disappeared in A-ZIF after 50 cycles (Fig. S26).

Supporting Info Page 36:

Fig. S26. ATR-IR spectra of A-ZIF (a) and AE-ZIF (b) between 0.85-1.55 V with different CV cycles (0, 20, 50 and 100).

3. Single-ligand A-ZIF exhibited an interesting trend of OER activity in electrocatalysis, initially increasing and then decreasing. What are the detailed reasons for this outcome? Is it solely caused by phase transition, or are there other factors at play?

Response: Indeed, this is an interesting phenomenon. We also think that this behavior is mainly attributed to the dynamics of the step-wise phase transition. As discussed in our manuscript and previous studies², the tetrahedral Co sites in ZIF-67 is first oxidized into various types of Co(OH)₂ under certain potentials before reaching the higher-valence CoOOH, i.e. involving irreversible changes in valence state of Co from Co²⁺ to Co³⁺ and Co⁴⁺. The final Co³⁺ and Co⁴⁺ species likely serve as the real active sites, thus influence the electrocatalytic OER activity.

After prolonged amperometric test, the OER activity of A-ZIF decreases, likely due to two main reasons:

1. The continuous transformation of high-activity Co species to lower-activity Co species leads to an overall decrease in the collective activity of the Co sites, ultimately reducing the OER activity.
2. Prolonged potential application can cause a continuous decline in the intrinsic porosity of A-ZIF, which disrupts the porous structure and reduces the interaction opportunities between the electrolyte and the potential active sites, resulting in a sustained decrease in activity.

4. An in-situ stable (oxy)hydroxide layer was formed for AE-ZIF during the surface reconstruction process. Does the thickness of this layer affect the performance of your samples? Can you calculate the thickness of the layer formed at different potentials?

Response: We consider that the thickness of the in-situ formed (oxy)hydroxide layer does indeed affect the electrocatalytic activity of the samples to some extent. As indicated by the DFT calculations in **Figs. 5c** and **5d**, there is a synergistic interaction at the interface between ligand E and the (oxy)hydroxide layer, which can lower the OER energy barrier and exhibit a more concentrated electron density.

By adjusting the potential, electrocatalytic time, or electrolyte concentration, we can influence the extent of surface reconstruction of the ZIF particles to some degree. Quantification of its thickness at different potentials, however, is rather challenging. As mentioned before, the (oxy)hydroxide layer on each ZIF particle's surface often differs throughout the process. Therefore, to calculate the precise layer thickness at different potentials, we would require advanced in-situ electrochemical TEM support on lots of samples. However, the presence of a liquid electrolyte in the electrochemical process complicates in-situ electrochemical TEM ⁵, and most MOF struggle to maintain stable structures under prolonged electron beam irradiation exposure ⁶, which makes electron microscopy testing time-consuming and inherently challenging.

Fig. 7: c, Charge density difference of the CoOOH incorporating ligands A and E models. **d**, Free energies diagram of the CoOOH incorporating ligands A and E models.

5. There are Co(OH)₂ and CoOOH in the (oxy)hydroxide layer for AE-ZIF and AD-ZIF, which were not prominently observed in XRD and XPS but were detected in Raman spectroscopy. More detailed reasons for this should be discussed.

Response: This is a good question. The main reason is that XRD and XPS are ex-situ measurements, and cannot directly capture the high-valence metal oxide signals in ultrathin

layers at different potential windows as sensitively as in-situ electrochemical Raman spectroscopy.

More details for ex-situ XRD and XPS, we use the test means of scraping off a small amount of the ZIF powder from the FTO glass electrode after the reaction. For uncontrollable factors, insufficient particle quantity and excessively thin outer layers contributed to the difficult to detect significant $\text{Co}(\text{OH})_2$ and CoOOH signals by ex-situ XRD and XPS measurements.

Generally, Raman is more sensitive for ultra-thin films, using our in-situ electrochemical cell (**Fig. 5a**), the laser can be directly guided onto the ZIF particles on the FTO glass, allowing real-time monitoring of the surface reconstruction on the ZIF surface. Additionally, it is easier to modify the electrocatalytic parameters and conditions in the Raman measurement, such as potential, cycle number, and duration time. This enables a better verification of the phase transition occurring on the surface of AE-ZIF, with the electro-oxidation forming high-valence $\text{Co}(\text{OH})_2$ and CoOOH .

Fig. 5: a, Cell schematic diagram in the three-electrode system of in-situ electrochemical Raman spectroscopy and UV-vis absorption spectroscopy.

6. Figure 1e shows the ^1H NMR spectrum of ZIFs. The main text should specify the solvent used for digestion, d_4 -acetic acid?

Response: We thank the referee for the suggestion, it was added.

Page 6: All samples were digested using d_4 -acetic acid and then tested by ^1H NMR.

7. In Figure 3 of the XPS data, it is necessary to clearly label which peaks represent Co-N, C-N and other relevant bonds. Similarly for N 1s. Specifically, indicate with distinct colors or labels: blue for Co-N and red for C-N. Additionally, the potential window is too small to read, so the font needs to be adjusted up.

Response: Your suggestions are very reasonable and clear. We have added distinct annotations in **Figs. 3c** and **3d** to clearly indicate the meaning of each characteristic peak. Additionally, we have increased the font size for the potential window labels to make it easier for readers to understand.

Page 10: Figs. 3c and **d** summarize the results from X-ray photoelectron spectroscopy (XPS) analysis of the as-prepared ZIFs before and after electrocatalysis. The Co 2p_{3/2} spectra of A-ZIF show two peaks before the reaction, corresponding to Co²⁺ coordinated to N on the ligand (782.09 eV, red) and satellite peak (787.79 eV, blue)^{7,8}. After 100 CV cycles in the medium or high potential ranges, a new distinct peak appeared at 779.14 eV (green) typically ascribed to Co³⁺ generated from Co²⁺ oxidation^{9,10,11}.

Page 11: This is further supported by the vanishing N 1s signals corresponding to C-N (400.15 eV, blue) and Co-N (398.85 eV, red), respectively.

Fig. 3: Structure evolution of ZIFs after electrocatalysis. **a**, XRD patterns of A-ZIF and AE-ZIF with different CV cycles (25, 50 and 100) at 0.85-1.55 V, with 100 CV cycles at different potential windows (Low: 0.85-1.05 V, Medi: 0.85-1.30 V, High: 0.85-1.55 V). **b**, XRD patterns of A-ZIF, AB-ZIF, AC-ZIF, AD-ZIF and AE-ZIF with 100 CV cycles at 0.85-1.55 V. The orange dot is ZIF signal, the purple square is Co(OH)₂ signal, and the pink plum is CoOOH signal. **c**, **d**, Co 2p_{3/2} and N 1s of XPS spectra for A-ZIF (**c**) and AE-ZIF (**d**) with 100 CV cycles at different potential window (Low: 0.85-1.05 V, Medi: 0.85-1.30 V, High: 0.85-1.55 V), data from **Fig. S17**.

Supporting Info Page 23: Moreover, two signals corresponding to Co-N (red) and C-N (blue) were retained in the N 1s spectra¹⁰, as depicted in **Figs. S19** and **S20**.

8. In the schematic diagrams of DFT calculations shown in Figures 5c and 5d, it is necessary to directly annotate the colors representing different atoms. Additionally, they are too close to other figures; it is recommended to separate them to avoid confusion.

Response: We fully agree with your suggestion. We have added annotations indicating the colors representing different atoms in **Fig. 5c** and **5d** and have also increased the spacing between them to enhance clarity and facilitate better understanding of the ZIF structure for readers.

Page 21:

Fig. 7: DFT simulations and mechanisms diagram of as-prepared ZIFs reconstruction. **a, b**, Optimized model structures **(a)** and calculated total DOS **(b)** of A-ZIF, AB-ZIF, AC-ZIF, AD-ZIF and AE-ZIF. **c**, Charge density difference of the CoOOH incorporating ligands A and E models. **d**, Free energies diagram of the CoOOH incorporating ligands A and E models. More details of the single-cell models and the free energy calculation steps are provided in **Figs. S46-54**. **e, f**, Schematic diagram of the complete reconstruction mechanism **(e)** and surface reconstruction mechanism **(f)** in (photo)electrocatalytic reaction.

9. The authors are recommended to read the following manuscripts (Appl. Catal. B-Environ. Energy 2024, 350: 123925; Chem. Eng. J. 2023, 455: 140601) for improving the manuscript.

Response: We appreciate your suggestion. We have reviewed the aforementioned papers and cited them as Refs. 9 and 11 in the revised manuscript to improve its quality.

Changes:

The following citations have been added to the list:

9. Dong P, Pan J, Zhang L, Yang X-L, Xie M-H, Zhang J. Regulation of electron delocalization between flower-like (Co, Ni)-MOF array and WO₃/W photoanode for effective photoelectrochemical water splitting. *Appl. Catal. B-Environ. Energy* **350**, 123925 (2024).
11. Tayyab M, *et al.* A new breakthrough in photocatalytic hydrogen evolution by amorphous and chalcogenide enriched cocatalysts. *Chem Eng J* **455**, 140601 (2023).

Reviewer #3

General comment. Authors synthesized four distinct mixed-ligand versions of zeolitic imidazolate framework-67 (ZIF-67) and conducted a comprehensive investigation into the structural evolution and degradation processes during electrocatalytic oxygen evolution reactions. The key point, surface reconstruction to form a protective cobalt (oxy)hydroxide layer, boosts both OER activity and stability. This scope is interesting and fundamental, and the whole story sounds good. This work is well organized and the results are important and convincing. Therefore, I would like to recommend its publication on Nature Communications after following minor questions being responded.

Response: We really appreciate this comment, which is very helpful to improve our manuscript. We have answered all the questions you asked below and hope that the changes satisfy you.

1. Both AC and AD samples show enhanced stability, however, their activity is lower than AE. What is the reason for this difference? Is their performance determined by the initially mixed new ligands?

Response: This is a very valuable question. As you mentioned, the difference in electrocatalytic activity is mainly due to the initial mixing of ligands. Ligand E both has additional -NH_2 groups and π - π stacking aromatic carbon rings, which provide superior electronic absorption and transfer capabilities compared to ligands C and D. This has been demonstrated through conductivity tests and DOS simulations (**Figs. 7b** and **S14**). Moreover, the optimal synergistic effect between ligand E and CoOOH effectively reduces the energy barrier for the rate-determining step (RDS, O^* oxidized to OOH^* ; in **Fig. 7d**), and energy difference (ΔE) between ϵ_d (Co band center) and ϵ_p (N band center). These advantages resulted in higher OER activity and strongest durability of AE-ZIF with mixed ligand E compared to AC-ZIF and AD-ZIF samples with mixed ligands C and D.

Certainly, to better clarify our point above, we have made the following revisions to the manuscript's description, ensuring that readers can understand it more easily.

Page 19: In summary, the ligand E mixed in AE-ZIF play a vital role enhancing OER activity, featuring both has additional -NH_2 groups and π - π stacking aromatic carbon rings, which provide superior electronic absorption and transfer capabilities compared to ligands B, C and D.

Fig. S14. Electrical conductivity (log scale) of A-ZIF (a), AB-ZIF (b), AC-ZIF (c), AD-ZIF (d) and AE-ZIF (e) on FTO glass between 0.85-1.55 V with different CV cycles (0, 10, 20, 30, 40, 50, 60, 70, 80, 90 and 100).

Fig. 7: a, b, Optimized model structures (a) and calculated total DOS (b) of A-ZIF, AB-ZIF, AC-ZIF, AD-ZIF and AE-ZIF. **c,** Charge density difference of the CoOOH incorporating ligands A and E models. **d,** Free energies diagram of the CoOOH incorporating ligands A and E models. More details of the single-cell models and the free energy calculation steps are provided in **Figs. S46-54.**

2. In Figure 2, the O₂ evolution curves show a significant upward trend within the initial time. Is this caused by reconstruction or the accumulation of O₂ in the device?

Response: As you mentioned, unlike the increase in current density of the samples within the first two hours, the upward trend in the O₂ evolution curves during the initial period is due to the natural slow accumulation in O₂ content within the cell reactor we constructed. This phenomenon is quite common; even with previously reported stable traditional semiconductor materials tested using the same device^{12,13}, a similar accumulation of O₂ release occurs during the process. Therefore, we cannot use the O₂ evolution rate curve to judge the reconstruction mechanism of ZIF compared to the current density curve. This accumulation process is entirely determined by the volume of the reactor and the flow rate of the inert gas.

3. The characterization, which is conducted to explore the reconstruction process, shows a decreased signal intensity for the characteristic peaks. What is the reason? Is there structural degradation?

Response: Thank you for the question about structural reconstruction. If you are referring to the XRD and XPS characteristic peaks of AE-ZIF in **Fig. 3**, they do not show a significant decrease. However, we understand your concern regarding the weakening of the characteristic peaks of the stable AE-ZIF sample observed in the in-situ UV-vis and Raman analysis over time (**Fig. 4**). This is a very common phenomenon in in-situ electrochemical testing³, primarily because at certain potentials, small O₂ bubbles are generated on the sample electrode (FTO glass). These bubbles can affect the detector device's ability to receive laser light, thereby impacting the final characteristic signals. However, this does not indicate that the tested samples have undergone structural degradation. Observing the XRD and XPS characteristic peaks of AE-ZIF after the OER reaction in **Fig. 3**, there is no significant decrease in intensity. Additionally, the TEM morphology of AE-ZIF is largely unchanged, retaining the inherent dodecahedral structure (**Fig. 4**). Therefore, we believe that our most stable AE-ZIF does not exhibit the significant structural degradation in the electrochemical OER under a certain potential window.

4. In Figure 1, the authors should show the full IR spectrum and note which signals originate from the ligands and which are from the metal-ligand bonding.

Response: We agree with the reviewer's viewpoint. We have provided the complete IR spectrum in **Fig. S3** of the Supplementary Information. To aid in understanding, we have added annotations for the ligand peaks in **Fig. 1d** of the main manuscript. Specifically, peaks below 505 cm^{-1} are characteristic of Co-N bonds between the ligand nitrogen and the metal cobalt. The cumulative peaks in the $505\text{ to }1800\text{ cm}^{-1}$ range originate from the stretching vibrations of the ligand structures.

Page 7:

Fig. 1: Structural characterization of Ligand-engineered ZIFs. a, b, Schematic diagram of the various ligands A-E mixed in LE-ZIFs (a) and unit cell of each LE-ZIFs (b). c, d, e, XRD patterns (c), ATR-IR (d) and ^1H NMR spectroscopy (e) of A-ZIF, AB-ZIF, AC-ZIF, AD-ZIF and AE-ZIF, more details in **Figs. S1, S3** and **S21-S25**. All samples were digested using d^4 -acetic acid and then tested by ^1H NMR. The numbers in **Fig. 1e** are for H at the corresponding positions on the ligand in **Fig. 1a**.

5. A line scan analysis is suggested to conduct to show the core-shell structure of AE-ZIF in Figure 4.

Response: Thank you for your suggestion. We have added an additional line scan in **Fig. 4d**. At the edges of the particles, the content of Co and O elements is significantly higher than that of N and C, which better demonstrates that the surface of the AE-ZIF particles has been reconstructed into a high-valence cobalt (oxy)hydroxide layer, resembling a core-shell structure.

Page 14:

Fig. 4: Morphology evolution of AE-ZIF after electrocatalysis. **a, d**, TEM image of single-particle AE-ZIF before **(a)** and after 12-hour amperometry **(d)**. **b, e**, Schematic diagrams and enlarged TEM images of AE-ZIF edge before **(b)** and after 12-hour amperometry **(e)**, the lower corresponding FFT patterns. **c, f**, Elemental mapping of single-particle AE-ZIF before **(c)** and after **(f)** 12-hour amperometry showing the Co, O, N and C distribution.

6. Many of the annotations in Figure 5 are too small to read.

Response: We apologize for the difficulty in reading the annotations. We have adjusted and enlarged all annotations in **Fig. 5** to make them more accessible and easier for readers to understand.

Fig. 5: Structure evolution of ZIFs by in-situ spectro-electrochemical analysis. **a**, Cell schematic diagram in the three-electrode system of in-situ electrochemical Raman spectroscopy and UV-vis absorption spectroscopy. **b**, In-situ electrochemical UV-Vis absorption study of A-ZIF and AE-ZIF with 100 CV cycles at 0.85-1.55 V. **c**, In-situ electrochemical Raman study and 2D contour plot of A-ZIF and AE-ZIF at various applied potentials from 1.20 to 1.65 V. Multistep amperometry was employed with a scanning interval of 5 minutes for each potential increase. **d**, In-situ electrochemical Raman study and 2D contour plots of A-ZIF and AE-ZIF with 100 CV cycles at 0.85-1.55 V. Orange dot is ZIF signal, purple square is Co(OH)₂ signal, and pink plum is CoOOH signal.

7. The overpotential of catalyst at 10 mA cm⁻² is an important reference to evaluate the activity. The authors could present this basic data.

Response: We agree with your suggestion that the overpotential of a catalyst at 10 mA cm⁻² is a common and important metric for evaluating catalytic activity. We have provided the overpotential data for our ZIF samples and conversion formula for the current density unit

from A g^{-1} to mA cm^{-2} in Supporting Information. However, we need to emphasize that all electrocatalytic experiments in our manuscript were conducted with ZIF samples drop-cast on FTO glass as the electrode. Thus, directly comparing our results with those using GCE or nickel foam electrodes would not highlight the advantages of our study. Under the same experimental conditions, the surface-reconstructed AE-ZIF exhibited an order of magnitude higher OER activity and better stability compared to single-ligand A-ZIF, demonstrating the potential of our ligand engineering strategy in OER electrocatalysis. Importantly, this manuscript focuses on the reconstruction phenomena of ligand-engineered ZIFs and the growth mechanism of the cobalt (oxy)hydroxide layer during electrocatalysis, rather than solely on performance enhancement.

Supporting Info Page 73:

Conversion formula for the current density unit of our samples from A g^{-1} to mA cm^{-2} is: $3.33 \text{ A g}^{-1} = 10 \text{ mA cm}^{-2}$. Using this formula, the overpotential for AE-ZIF at 10 mA cm^{-2} is 1.53 V, while A-ZIF only reaches 2 mA cm^{-2} at the same potential.

7. How do the authors construct the ZIF models with mixed ligands in DFT calculations? It is suggested to provide the input files for each ZIF structure considered in this manuscript.

Response: When calculating the ZIF models with different ligands, the process involved replacing the original ligand A, which has $-\text{CH}_3$ (methyl) groups, in the ZIF structure with ligands B-E, as depicted in **Fig. 7a**. This modification was carried out while maintaining the rest of the ZIF framework unchanged to ensure consistency in the comparison of results. For the OER barrier calculations, the hydroxide oxide was directly linked to ligand A/E to assess its impact on the reaction (**Fig. S53**). In contrast, the comparison sample was prepared with the hydroxide oxide without any connection to ligands, ensuring that only the effect of the ligand was evaluated. Both models share the same reaction sites, as illustrated in **Fig. 7d**, allowing for a direct comparison of the influence of ligands on the OER barrier.

The input files (CIF) and model data for each optimized ZIF structure discussed in this study have been provided in the attached compressed file (Optimized ZIF models) to facilitate replication of the calculations.

Fig. S53. Calculation steps for OER free energy of A-CoOOH (a) and E-CoOOH (b). Dark purple for Co, blue for N, gray for C and white for H.

7. The following reports are for the references to improve the manuscript: Appl. Catal. B-Environ. 2024, 353, 124096; J. Colloid Interface Sci. 2024, 653: 380; Angew. Chem. Int. Ed. 2022, 134: e202214794.

Response: We appreciate the recommended papers. We have reviewed the papers mentioned above and cited them in the corresponding parts of the main text as Refs. 17, 51 and 53 in the revised manuscript.

Changes:

The following citations have been added to the list:

17. Zhang L, *et al.* Self-Reconstructed Metal-Organic Framework Heterojunction for Switchable Oxygen Evolution Reaction. *Angew Chem Int Ed* **61**, e202214794 (2022).
51. Li X, *et al.* Boosting photoelectrocatalytic oxygen evolution activity of BiVO₄ photoanodes via caffeic acid bridged to NiFeOOH. *Appl. Catal. B-Environ. Energy*

- 353**, 124096 (2024).
53. Chen M, *et al.* Metalloporphyrin based MOF-545 coupled with solid solution $Zn_xCd_{1-x}S$ for efficient photocatalytic hydrogen production. *J Colloid Interface Sci* **653**, 380-389 (2024).

References

1. Wei C, Sun S, Mandler D, Wang X, Qiao SZ, Xu ZJ. Approaches for measuring the surface areas of metal oxide electrocatalysts for determining their intrinsic electrocatalytic activity. *Chem Soc Rev* **48**, 2518-2534 (2019).
2. Zheng W, Liu M, Lee LYS. Electrochemical Instability of Metal–Organic Frameworks: In Situ Spectroelectrochemical Investigation of the Real Active Sites. *ACS Catalysis* **10**, 81-92 (2020).
3. Cheng F, *et al.* Accelerated water activation and stabilized metal-organic framework via constructing triangular active-regions for ampere-level current density hydrogen production. *Nat Commun* **13**, 6486 (2022).
4. Tran TTN, *et al.* Dopant-Induced Charge Redistribution on the 3D Sponge-like Hierarchical Structure of Quaternary Metal Phosphides Nanosheet Arrays Derived from Metal–Organic Frameworks for Natural Seawater Splitting. *ACS Appl Mater Interfaces*, (2024).
5. Qu J, Sui M, Li R. Recent advances in in-situ transmission electron microscopy techniques for heterogeneous catalysis. *iScience* **26**, 107072 (2023).
6. Liu L, Zhang D, Zhu Y, Han Y. Bulk and local structures of metal–organic frameworks unravelled by high-resolution electron microscopy. *Communications Chemistry* **3**, 99 (2020).
7. Xue Z, *et al.* Missing-linker metal-organic frameworks for oxygen evolution reaction. *Nat Commun* **10**, 5048 (2019).
8. Chen H, *et al.* Preparation of reduced graphite oxide loaded with cobalt(II) and nitrogen co-doped carbon polyhedrons from a metal-organic framework (type ZIF-67), and its application to electrochemical determination of metronidazole. *Microchimica Acta* **186**, 623 (2019).
9. Yang J, Liu H, Martens WN, Frost RL. Synthesis and Characterization of Cobalt Hydroxide, Cobalt Oxyhydroxide, and Cobalt Oxide Nanodiscs. *J Phys Chem C* **114**, 111-119 (2010).
10. Zhu R, *et al.* Quasi-ZIF-67 for Boosted Oxygen Evolution Reaction Catalytic Activity via a Low Temperature Calcination. *ACS Appl Mater Interfaces* **12**, 25037-25041 (2020).
11. Xue S, *et al.* In situ constructing Co/Co-Ox/Co-Nx diverse active sites on hollow porous carbon spheres derived from Co-MOF for efficient bifunctional electrocatalysis in rechargeable Zn-air. *Mater Today Phys* **37**, 101209 (2023).
12. Schubert JS, *et al.* Nature of the Active Ni State for Photocatalytic Hydrogen

- Generation. *Adv Mater Interfaces* **11**, 2300695 (2024).
13. Ayala P, *et al.* The Emergence of 2D Building Units in Metal-Organic Frameworks for Photocatalytic Hydrogen Evolution: A Case Study with COK-47. *Advanced Energy Materials* **13**, 2300961 (2023).